# Solving the MCM paradox by visualizing the scaffold of CMG helicase at active replisomes

Hana Polasek-Sedlackova [1,2] ✉, Thomas C. R. Miller [3], Jana Krejci [2], Maj-Britt Rask[1] & Jiri Lukas [1] ✉

Genome duplication is safeguarded by constantly adjusting the activity of the replicative CMG (CDC45-MCM2-7-GINS) helicase. However, minichromosome maintenance proteins (MCMs)—the structural core of the CMG helicase—have never been visualized at sites of DNA synthesis inside a cell (the so-called MCM paradox). Here, we solve this conundrum by showing that anti-MCM antibodies primarily detect inactive MCMs. Upon conversion of inactive MCMs to CMGs, factors that are required for replisome activity bind to the MCM scaffold and block MCM antibody binding sites. Tagging of endogenous MCMs by CRISPR-Cas9 bypasses this steric hindrance and enables MCM visualization at active replisomes. Thus, by defining conditions for detecting the structural core of the replicative CMG helicase, our results explain the MCM paradox, provide visual proof that MCMs are an integral part of active replisomes in vivo, and enable the investigation of replication dynamics in living cells exposed to a constantly changing environment.

Accurate and complete DNA replication is fundamental to cell proliferation. In the late mitosis and G1 phase, origins of replication are licensed-to-replicate by loading pairs of MCM2-7 complexes onto DNA to form inactive MCM double hexamers (MCM-DHs)[1–3]. MCM-DHs are converted to active replicative CMG (CDC45-MCM2-7-GINS) helicases during origin firing at the onset of the S phase[4]. Two seemingly paradoxical questions have been asked about MCMs since the discovery of their crucial role in DNA replication[5,6]. First, why do cells load a huge excess of MCMs onto chromatin if only 5−10% are used as active CMG helicases? Second, why do eukaryotic MCMs not localize to active replication factories (RFs) visualized by immunofluorescence (IF)?[7–10]. In recent years, we and others have provided an explanation for the first MCM paradox. The need to load an excess of inactive MCMs is underlined by their roles in adjusting the physiological speed of DNA replication[11], as well as serving as a backup pool of origins that can be activated to rescue stalled or collapsed replication forks[12,13]. In contrast, the second MCM paradox remains largely unaddressed, with few potential explanations proposed in the past decades. One previous model suggested that the replicative helicase is located at distance from active replication sites and pumps ssDNA towards them[14]. Other studies proposed technical explanations for the failure to visualize MCMs in RFs. For instance, it was hypothesized that the large fraction of inactive chromatin-bound MCMs might hamper immunodetection of the minor fraction of active MCMs at replication forks[8–10,15,16]. Similarly, it was also suggested that the local fluorescence signal of active MCMs at replication forks might be reduced beyond the detection threshold by local chromatin decondensation[17]. However, none of these hypotheses have been proven and it thus remains unclear whether the replisome structure in a cellular context is different from the current models based on biochemical and structural studies or whether the previous imaging approaches simply failed to visualize active MCMs at RFs. The paucity of rigorous explanation of this phenomenon hampers further progress in studying replication dynamics in physiological settings and the key motivation of this manuscript was to remove this obstacle by asking two intrinsically connected questions: Is CMG helicase an integral component of functional replisomes in cells as described by various structural studies? And, if so, why did previous attempts fail to visualize CMG or particularly MCMs at active RFs?

[1]Protein Signaling Program, Novo Nordisk Foundation Center for Protein Research, Faculty of Health and Medical Sciences, University of Copenhagen, Copenhagen, Denmark. [2]Department of Cell Biology and Epigenetics, Institute of Biophysics, Czech Academy of Sciences, Brno, Czech Republic. [3]Center for Chromosome Stability, Department of Cellular and Molecular Medicine, Faculty of Health and Medical Sciences, University of Copenhagen, Copenhagen, Denmark. ✉e-mail: polasek-sedlackova@ibp.cz; jiri.lukas@cpr.ku.dk

## Results and discussion

To address the first question[14], we applied IF to interrogate the colocalization of CMG helicase with replication sites. We generated U2OS cells expressing endogenously GFP-tagged CDC45 using CRISPR-Cas9 genome editing to monitor CMG dynamics without adverse effects of protein overexpression[11]. The CDC45-GFP cell line was thoroughly tested to validate homozygous tagging of all CDC45 alleles (Supplementary Fig. 1a). Using quantitative image-based cytometry of large cell populations (QIBC)[18], we confirmed that chromatin binding of CDC45 during the S phase is confined to actively replicating DNA marked by a short pulse of EdU (Supplementary Fig. 1b, c). Notably, QIBC analysis of chromatin bound CDC45 enabled us to define individual S phase stages to perform a more refined analysis of replication dynamics at individual replisomes, which are clearly discernible in late S-phase fractions (Supplementary Fig. 1b, c). Consistent with it being a core replisome component, CDC45 showed high colocalization with core components of active RFs marked by PCNA or EdU (Fig. 1a, b). Similarly, endogenously tagged leading and lagging strand polymerases POLE1 and POLD1, and immunostained fork speed accelerator TIMELESS strongly localize to replication sites (Fig. 1b, Supplementary Fig. 1d–f). Collectively these data generate conditions to visualize CMG in the context of functional replisomes in cells and show that endogenously tagged CDC45 is a robust surrogate of RFs undergoing active DNA synthesis.

To visualize MCMs within the CDC45-marked RFs, we selected a panel of MCM antibodies and thoroughly analyzed their binding to chromatin during the cell cycle by immunostaining and QIBC (Supplementary Fig. 2a). In contrast to other replisome components, individual subunits of the MCM2-7 complex show gradual dissociation from chromatin upon initiation of DNA replication and throughout S phase (Fig. 1c, Supplementary Fig. 2a, Supplementary Fig. 3a). This MCM dynamics is expected and reflects replication-coupled removal of the excess of inactive MCMs bound to chromatin (Fig. 1d). Consistent with previous studies, confocal microscopy failed to reveal a clear fraction of MCMs at active replisomes. In fact, we find very low colocalization of MCMs with CDC45 or PCNA even in the late S phase (Fig. 1e, f, Supplementary 3b), where DNA synthesis occurs in discrete RFs and the bulk of inactive MCMs have already been removed from chromatin (Fig. 1d). The inability to detect immunostained MCMs at active replisomes was further validated by increasing the stringency of the detergent pre-extraction step before immunostaining (Supplementary Fig. 3c, d). This striking observation that CDC45, a direct component of CMG, is rarely colocalized with immunostained MCMs led us to revisit a previous explanation of this classical cytological manifestation of the MCM paradox, namely that the presence of a large fraction of inactive origins masks a small active fraction of CMG-associated MCMs[15–17]. To this end, we inhibited ATR, the key upstream regulator of the S-phase checkpoint, whose inhibition converts inactive MCMs to CMGs[18]. We reasoned that such conditions of excessive origin activity would reduce the fraction of inactive MCMs and reveal the presence of their active counterparts at RFs. Strikingly, however, in contrast to replisome components such as CDC45 or TIMELESS, whose chromatin binding increased approximately four-fold in the presence of ATR inhibitor (Fig. 2a, Supplementary Fig. 4a), immunostained MCMs were even less abundant on actively replicating chromatin (Fig. 2b, c, Supplementary Fig. 4b). This was particularly visible in S5 and S6 fractions where DNA replication can be tracked in individual RFs with high accuracy (insets in Fig. 2c; and see Supplementary Fig 4b). Consistent with these findings, we observed decreased colocalization of MCMs with CDC45 or PCNA when analyzed by confocal microscopy at individual RFs following ATR inhibition (Fig. 2d–f, Supplementary Fig. 5a, b). Interestingly, out of all MCM antibodies used in our study (12 in total), only MCM6 #2 showed higher detection on chromatin and increased colocalization with CDC45 and PCNA upon ATR inhibition (Fig. 2e, f, Supplementary Fig 4b, Supplementary

Fig. 5a, b). However, even in this case, the MCM6 #2 antibody detected only a fraction of RFs, as the Pearson correlation coefficient did not reach the maximal level compared to the CDC45 and PCNA colocalization (Fig. 2f, Supplementary Fig. 5b). Our observation that the MCM6 #2 antibody could partially visualize RFs was notable and will be discussed in detail in the following paragraphs.

The inability to efficiently detect MCMs in RFs by immunostaining even after increased origin firing led us to postulate that MCMs may become inaccessible to antibody binding after their conversion from inactive double hexamers to active replisomes. In support of this notion, structural comparison of an MCM double hexamer (*S. cerevisiae*) and the human core leading strand replisome[19,20] revealed that the accessibility of MCMs differs substantially in the two complexes (Fig. 2g, h). While in double hexamers the external surface of the hexameric MCM rings is freely accessible except for the N-terminal dimerization interfaces, the active MCMs in the replisome structure are largely occupied by multiple replisome components. For instance, MCM2 is partially hidden by interactions with CDC45, POLE, and CLASPIN, while MCM5 is buried under POLE, CDC45, and GINS subunits, which also interact with MCM3. An extensive interface between TIMELESS-TIPIN and the MCMs in the replisome could prevent antibody recognition of MCM7, MCM4, MCM6, and MCM2. Additionally, the MCM6-CLASPIN interaction could also prevent antibody recognition of active MCMs for IF. These results suggest that, in the context of the replisome, several MCM subunits may be difficult to reach, even by antibodies raised against full-length proteins. Notably, in addition to the steric occlusion for the replisome components described above, we propose that hitherto unmodelled regions of the replisome, additional replication factors[21] or posttranslational modifications of MCMs, such as those captured by mass-spectrometry approaches[22], could further curtail the accessibility of MCMs within the complex machinery of progressing replication forks in cells.

To bypass the confounding antibody binding hindrance, we applied CRISPR-Cas9 genome editing to endogenously tag MCM4 with a HaloTag in the CDC45-GFP cell line described earlier in this study. In control cells (DMSO treatment), comparison of immunostained and Halo-tagged MCMs by QIBC showed similar dynamics on chromatin, which is expected at the cell population level due to the excess of inactive MCMs (Fig. 3a, b, Supplementary Fig. 5c). In both cases, the fast dissociation of the bulk of MCMs can be explained by rapid eviction of inactive MCMs by progressing replication forks[11]. However, upon increasing the pool of active replication forks via ATR inhibition, the opposite behavior of immunostained and endogenously tagged MCMs was observed. While immunostained MCM4 and 7 disappeared from S-phase chromatin much faster, the signal associated with endogenous MCM4-Halo increased in comparison to the control cells (Fig. 3a, b, Supplementary Fig. 5c). We hypothesized that the increase in levels of endogenously tagged MCMs on chromatin reflects an increase in CMGs that, once active, remain on DNA until they meet with a replisome coming from the opposite direction[23,24]. Similar differences in the dynamics of immunostained and endogenously tagged MCMs on chromatin during the S phase were recapitulated with MCM2-GFP and corresponding MCM2 antibody (Supplementary Fig. 5d). Importantly, these findings obtained by population-based QIBC are fully consistent with single-cell confocal imaging, where endogenously tagged MCMs (but not corresponding immunostained MCMs) strongly co-localized with active RFs marked by CDC45 (Fig. 3c, d) or PCNA (Supplementary Fig. 6a). To demonstrate the general implications of our results, similar dynamics of endogenously tagged and immunostained MCMs was reproduced in RPE-1 cells (Supplementary Fig. 6b). Interestingly, the partial detection of active MCMs in RFs by the MCM6 #2 antibody described earlier (see Fig. 2e, f, Supplementary Fig 4b, Supplementary Fig. 5a, b) indicated that this antibody may be able to recognize MCMs after CMG maturation. However, both QIBC and colocalization analysis consistently support the

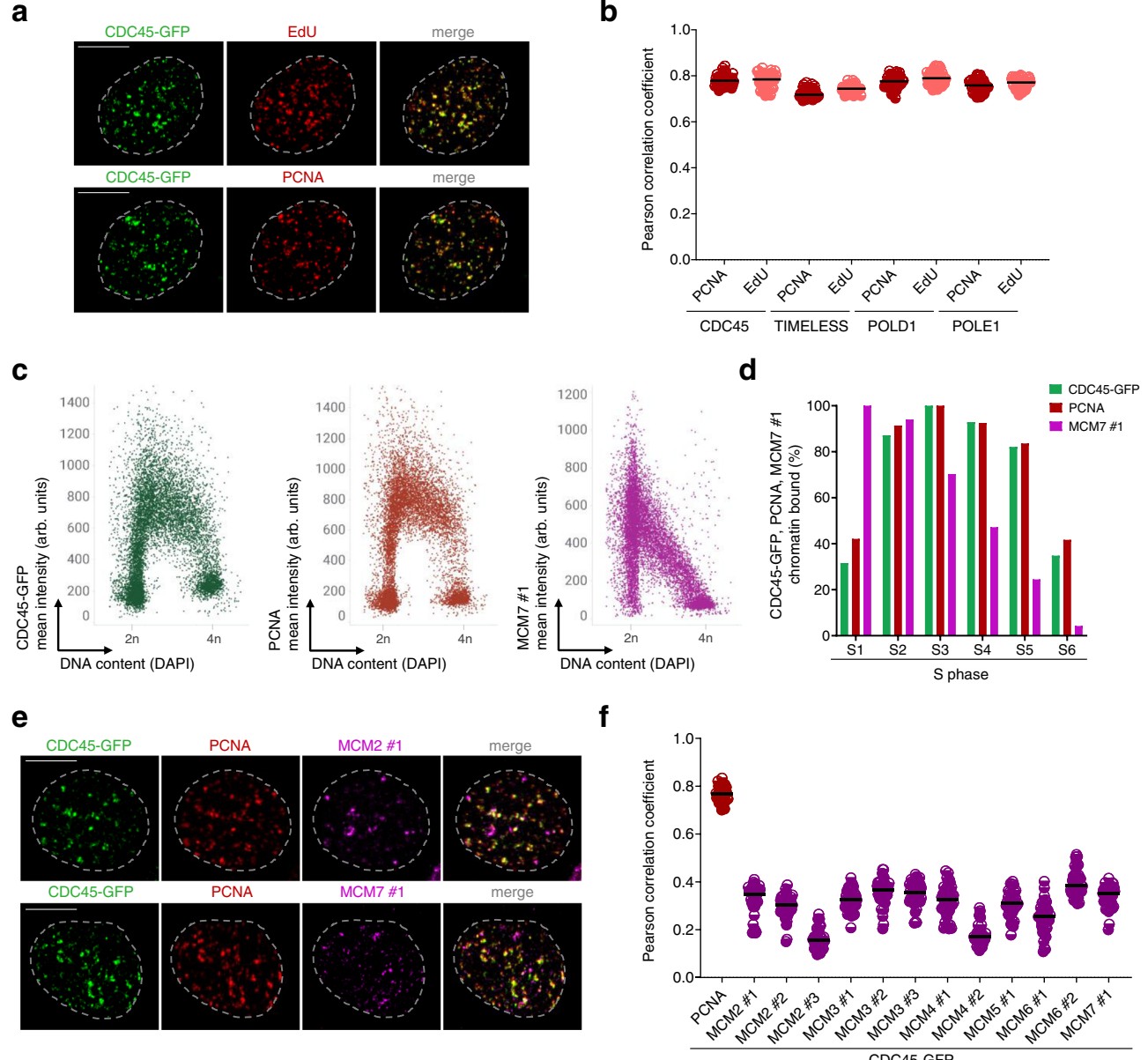

**Fig. 1 | CDC45, a component of CMG, marks active replication factories but shows low colocalization with MCMs by IF. a** Representative maximum intensity projection (MIP) confocal images of U2OS cells expressing endogenously tagged CDC45-GFP. Cells were pulse-labeled with EdU and counterstained for chromatin-bound PCNA to detect RFs in S5 stage of S phase sub-stratified as in (Supplementary Fig. 1b). Scale bar, 10 μm. **b** Colocalization analysis of endogenously tagged CDC45-GFP, POLD1-GFP, POLE1-GFP and immunostained TIMELESS with RFs visualized by PCNA antibody or EdU in S5 stage of S phase. Horizontal lines represent medians; *n* = 40 cells per condition. **c** QIBC of CDC45-GFP cells immunostained for chromatin bound GFP, PCNA and MCM7. Nuclear DNA was counterstained by DAPI; *n* ≈ 10,000 cells per condition; arb. (arbitrary) units. **d** Quantification of QIBC plots in **c** throughout S phase based on CDC45-GFP and DAPI intensities (see Supplementary Fig. 1b for S phase sub-stratification). Each bar indicates median of mean intensity normalized to 100 percent with respect to CDC45-GFP or PCNA intensities in the S3 stage of S phase or to MCM7 intensity in the S1 stage of S phase; *n* ≈ 10,000 cells per condition. **e** Representative MIP confocal images of chromatin-bound CDC45-GFP, PCNA and MCM2 or MCM7 detected by indicated antibodies in S5 stage of S phase sub-stratified as in (Supplementary Fig. 1b). Scale bar, 10 μm. **f** Colocalization analysis of immunostained MCMs with RFs visualized by endogenously tagged CDC45- in S5 stage of S phase. Horizontal lines are medians; *n* = 40 cells per condition. Source data are provided as a Source data file.

conclusion that in contrast to endogenously tagged MCMs, MCM6 detection with antibody #2 fails to detect all RFs (Supplementary Fig. 6c–e). This failure may reflect partial occlusion of MCM6 #2 antibody binding sites at some RFs by specific replisome binding factors. To rule out potential variations in the fluorescence signal intensities, all MCM antibodies and endogenously tagged MCM4 were compared under the same imaging condition (Supplementary Fig. 7a, b). This QIBC analysis demonstrates that the ability to detect some RFs with the MCM6 #2 antibody is not because of an outlier fluorescent signal but

most likely due to the unique accessibility of the MCM6 subunit after the formation of active replisomes.

Since CRISPR-Cas9 endogenous tagging of MCM subunits enables robust visualization of active MCMs at the replication fork, we wanted to understand why this had not been previously observed in mammalian cells with ectopically expressed GFP-tagged MCMs[17,25]. To this end, we performed colocalization analysis in previously described CHO cells[17] that conditionally overexpress MCM4-mEmerald using a tet-off promoter and stably express PCNA-RFP as a marker of replication

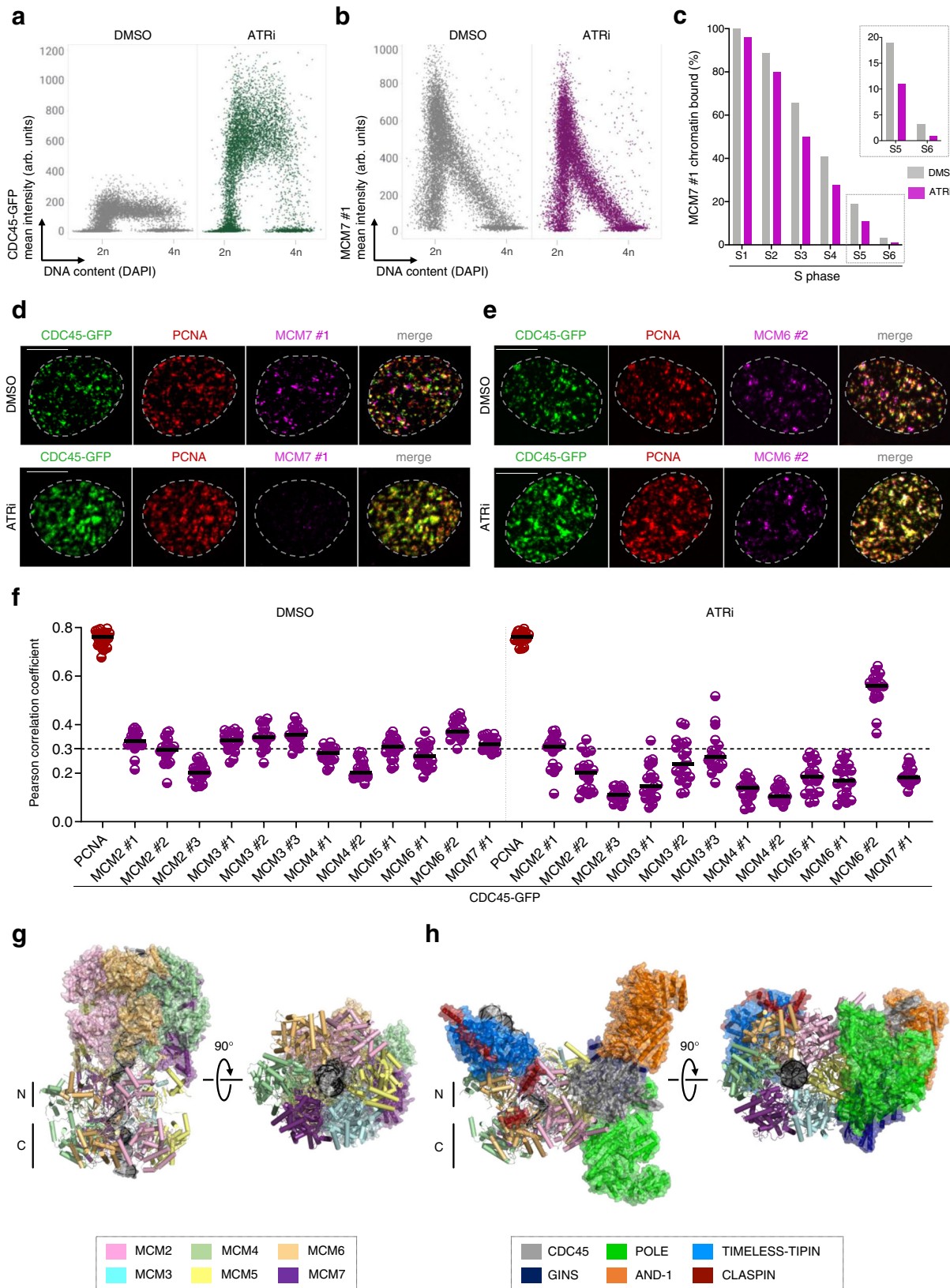

factories (Supplementary Fig. 7c, d). Similar to what we observed in IF-based assays using MCM antibodies (see for instance Fig. 2f, Supplementary Fig. 5b), we find a low level of colocalization between MCM4-mEmerald and PCNA-RFP-labeled RFs (Supplementary Fig. 7e), which is further reduced after increasing the abundance of active replisomes using an ATR inhibitor. In our previous work, we showed that the

cellular MCM equilibrium is maintained by tightly coupled recycling and biogenesis pathways[11]. Moreover, we found that parental MCMs inherited from the previous cell cycle were preferentially converted to CMG helicases, whilst newly synthesized 'nascent' MCMs were less efficiently converted to CMGs. Based on this finding, we hypothesized that the low level of colocalization between MCM4-mEmerald and

**Fig. 2 | Immunostained MCMs show decreased colocalization with RFs upon increased origin firing. a** QIBC of CDC45-GFP cells treated with ATR inhibitor (ATRi) and immunostained for chromatin-bound GFP. Nuclear DNA was counterstained by DAPI; $n \approx 10{,}000$ cells per condition; arb. (arbitrary) units. **b** QIBC of CDC45-GFP cells treated with ATR inhibitor (ATRi) and immunostained for chromatin-bound MCM7. Nuclear DNA was counterstained by DAPI; $n \approx 10{,}000$ cells per condition; arb. (arbitrary) units. **c** Quantification of QIBC plots in **a** and **b** throughout S phase based on CDC45-GFP and DAPI intensities (see Supplementary Fig. 1b for S phase sub-stratification). Each bar indicates median of mean intensity normalized to 100 percent with respect to MCM7 intensity in the S1 stage of S phase of untreated cells; $n \approx 10{,}000$ cells per condition. **d** Representative MIP confocal images of chromatin-bound CDC45-GFP, PCNA and MCM7 detected by indicated antibodies after ATR inhibitor treatment (ATRi) in S5 stage of S phase sub-stratified as in (Supplementary Fig. 1b). Scale bar, 10 µm. **e** Representative MIP confocal images of chromatin-bound CDC45-GFP, PCNA and MCM6 detected by indicated antibodies after ATR inhibitor treatment in S5 stage of S phase sub-stratified as in (Supplementary Fig. 1b). Scale bar, 10 µm. **f** Colocalization analysis of immunostained MCMs with RFs marked by endogenously tagged CDC45-GFP in S5 stage of S phase after ATR inhibitor treatment. Horizontal lines are medians; $n = 20$ cells per condition. The dashed horizontal line represents the average of the Pearson correlation coefficient for MCMs and CDC45-GFP colocalization in DMSO treatment. **g** Atomic model of *S. cerevisiae* MCM2-7 double hexamers loaded on DNA (PDB 6F0L (ref. [19])). **h** Atomic model of human core leading strand replisome on DNA (PDB 7PFO (ref. [20])). To highlight the distinct MCM interfaces that are occupied in the two complexes, a single MCM2-7 hexamer is displayed as a cartoon representation only, whilst all other MCMs and replication factors are displayed with a transparent surface. Images were generated in PyMOL (Schrödinger, LLC, v.2.5.2). Source data are provided as a Source data file.

PCNA-RFP-labeled RFs may reflect the fact that MCMs overexpressed from exogenous promoters are nascent MCMs that fail to efficiently form CMGs. Indeed, immunoprecipitation analysis revealed that while overexpressed MCM4-mEmerald can physically interact with other MCM subunits as previously reported[17], it has a reduced ability to interact with endogenous CDC45 relative to endogenously tagged MCM4 (Supplementary Fig. 7f). Consequently, detection of MCMs at RFs will be challenging in all studies that ectopically express MCMs without removing the cycling endogenous population. Collectively, we conclude that CRISPR-Cas9 endogenous tagging of MCM subunits effectively bypasses the steric inaccessibility of MCMs in fully assembled CMGs and might be the only robust and reliable way to visualize the active pool of MCMs in cells.

While the above experiments using endogenously tagged MCMs enabled us to visualize CMGs in RFs upon increased origin firing, we were naturally interested in whether the MCM scaffold can be detected also during the unperturbed S phase. Indeed, we repeatedly noticed that endogenously tagged MCMs show constantly higher Pearson correlation coefficient with RFs than MCMs detected by antibodies, even without ATR inhibitor (Fig. 3c, d, Supplementary Fig. 6a, b, d). To further explore whether the active pool of MCMs can be detected in RFs during normal DNA replication, we dissected the co-localization of endogenously tagged MCMs with CDC45 and immunostained MCMs throughout the whole S phase (Fig. 3e, f; see Supplementary Fig. 1b for S-phase sub-classification). In the early S phase, endogenously tagged and immunostained MCMs colocalized with each other, but in both cases, the Pearson correlation coefficient with RFs marked by CDC45 was relatively low due to the large fraction of inactive MCMs loaded on chromatin at this stage (Fig. 3e, f). Strikingly, however, once the bulk of inactive MCMs dissociated from chromatin in more advanced S-phase stages, colocalization between endogenously tagged and immunostained MCMs sharply declined. This was accompanied by opposite trajectories for endogenously tagged and immunostained MCMs with regards to their colocalization with CDC45, where the former markedly increased and the latter declined (Fig. 3e, f). Reassuringly, a very similar pattern of colocalization was obtained in the analysis of endogenously tagged MCM2-GFP and immunostained MCM2 at RFs marked by PCNA (Supplementary Fig. 8a, b). Hence, whereas the removal of inactive MCMs during S-phase progression is mirrored by a decrease of immunodetectable chromatin-bound MCMs, the corresponding unmasking of active CMGs (most pronounced in the late S phase) is marked by an increase of endogenously tagged MCMs at RFs. Importantly, we could validate that the latter are indeed sites of active DNA synthesis by reproducing the previous findings in cells exposed to a short pulse of EdU. Indeed, we could see a progressive increase in colocalization of endogenously tagged MCM4 with CDC45 and EdU, which peaked in the late S phase (Supplementary Fig. 8c, d). This finding not only validates the results obtained with CDC45 as a proxy for active RFs but also proves that the accumulation of endogenously tagged MCMs in the late S phase are progressing replication forks, not

terminated CMGs undergoing replisome disassembly. Of note, the Pearson correlation coefficient between CDC45 and EdU remains constantly high throughout the S phase, clearly indicating that DNA unwinding is tightly coupled with DNA synthesis (Supplementary Fig. 8c, d). Based on these data, we conclude that endogenous tagging of MCM subunits enables visualization of the entire pool of chromatin-bound MCMs, including previously undetectable CMG-associated fraction in active replication factories, while MCM antibodies are primarily detecting inactive MCMs.

To further support the hypothesis that replisome components directly shield the MCM antibody binding sites at fully assembled replication forks, we wanted to experimentally unmask active MCMs by the removal of CMG-associated accessory replisome components. To avoid extensive changes in the core replisome architecture, we focused on TIMELESS, CLASPIN, and AND-1, also known as replication protection complex (RPC), whose depletion for a short period of time only results in a moderate slowdown of replication forks and low levels of DNA damage[26]. Based on the structural analysis of the human core leading strand replisome (Fig. 2h)[20], we predicted that TIMELESS may directly shield regions of MCM7, MCM4, MCM6, and MCM2, whilst CLASPIN may shield epitopes in MCM6 and MCM2. AND-1, on the other hand, interacts with CMG via CDC45 and GINS and therefore represents an important negative control for our knockdown experiments.

To test that RPC is shielding various MCM subunits, we reduced levels of individual RPC components by siRNA (Supplementary Fig. 9a) and quantified the co-localization between CDC45 and immunostained MCMs in cells treated with ATR inhibitor. In support of our prediction, TIMELESS depletion indeed enhanced co-localization between CDC45 and immunostained MCM7 and MCM4 (Fig. 4a–c), but only moderately increased colocalization with the MCM5 subunit (Fig. 4d), which is on the opposite side of the MCM ring to the identified TIMELESS-CMG interaction interface (Fig. 2h). Similarly, CLASPIN knock-down led to increased detection of CDC45-marked RFs by MCM2 and MCM6 antibodies (Supplementary Fig. 9b–d), while colocalization between CDC45 and MCM7 antibodies was unaffected by CLASPIN removal (Supplementary Fig. 9e). Importantly, AND-1 downregulation had no effect on visualization of RFs by various MCM antibodies as the AND-1 interaction with CMG is mediated primarily through CDC45 and GINS and not MCMs (Supplementary Fig. 9g–j). Finally, co-localization of endogenously tagged MCM4-Halo with CDC45 remained constant in all cases of RPC depletion (Fig. 4a, e, Supplementary Fig. 9b, f, g, k) indicating that the observed effect of TIMELESS and CLASPIN depletion on detection of RFs is due to the steric occlusion of MCM antibody binding sites rather than an effect on DNA replication per se. Collectively, these data suggest that active MCMs representing the structural core of replication fork are coated by multiple replisome factors which limit antibody access in IF experiments. In agreement with previous structural studies, our approach supports the predicted position of individual RPC components in the human replisome structure formed directly in the cellular environment. Interestingly, the increased

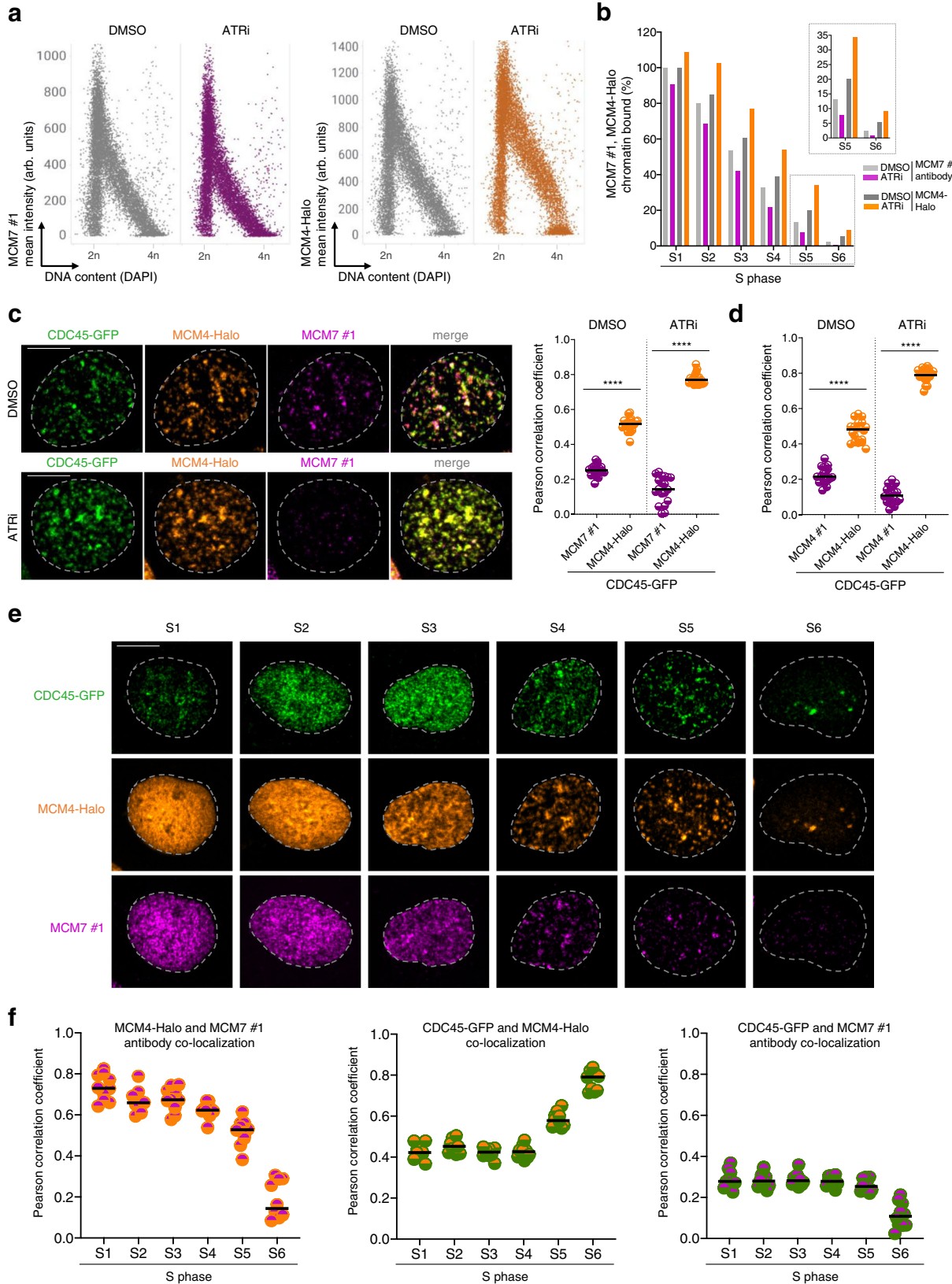

exposure of MCM antibody binding sites upon TIMELESS removal from active replisome raises the possibility that the inaccessibility of MCMs in the replisome may have additional biological implications. The removal of TIMELESS from active replisomes has been found to cause extensive fork rotation[27]. Combined with our previous work, which described TIMELESS dissociation from the replication fork in

response to mild replication stress[26], we suggest that modulation of MCM scaffold exposure could regulate the replication stress response by controlling access of fork-remodeling enzymes to the replication fork.

In conclusion, our study clarifies a longstanding discrepancy between the inability to detect MCMs in active replisomes by IF and

**Fig. 3 | Endogenous tagging of MCM subunits enables visualization of MCM scaffold at active replisomes. a** Left and right, QIBC of CDC45-GFP cells with endogenously tagged MCM4-Halo treated with ATR inhibitor (ATRi), pulsed with JF549 HaloTag ligand (200 nM; 20 min) and immunostained for MCM7. Nuclear DNA was counterstained by DAPI; $n \approx 10,000$ cells per condition; arb. (arbitrary) units. **b** Quantification of QIBC plots in **a** at different intervals of S phase sub-stratified as in (Supplementary Fig. 1b). Each bar indicates median of mean intensity normalized to 100 percent with respect to MCM7 or MCM-Halo intensities in the S1 stage of S phase in untreated cells; $n \approx 10,000$ cells per condition. **c** Left, representative MIP confocal images of chromatin-bound CDC45-GFP, MCM4-Halo and immunostained MCM7 in the S5 stage of S phase (see Supplementary Fig. 1b for S phase sub-stratification) during normal condition or after ATR inhibitor treatment. Scale bar, 10 μm. Right, colocalization analysis of endogenously tagged MCM4-Halo and immunostained MCM7 with RFs marked by endogenously tagged CDC45-GFP in the S5 stage of S phase during normal condition or after ATR inhibitor treatment. Lines are medians; $n = 20$ cells per condition; $P$ values were calculated by two-tailed unpaired $t$-test; ****$P > 0.0001$. **d** Colocalization analysis of endogenously tagged MCM4-Halo and immunostained MCM4 with CDC45-GFP in the S5 stage of S phase during normal condition or after ATR inhibitor treatment. Lines are medians; $n = 20$ cells per condition; $P$ values were calculated by two-tailed unpaired $t$-test; ****$P > 0.0001$. **e** Representative MIP confocal images of chromatin-bound CDC45-GFP, MCM4-Halo and immunostained MCM7 at the indicated S phase stages sub-stratified as in (Supplementary Fig. 1b). Scale bar, 10 μm. **f** Left, colocalization analysis of MCM4-Halo and immunostained MCM7 at the indicated S phase stages. Middle, colocalization analysis of CDC45-GFP and MCM4-Halo at the indicated S phase stages. Right, colocalization of CDC45-GFP and immunostained MCM7 at the indicated S phase stages. Lines are medians; $n = 10$ cells per condition. Source data are provided as a Source data file.

their essential role at the replication fork during genome duplication. Based on our findings, we conclude that anti-MCM antibodies primarily detect the highly abundant inactive pool of MCMs. Our analyses suggest that this is due to the fact that antibody binding sites become occluded by the recruitment of replisome components and are thus less accessible upon maturation of the chromatin-associated MCM rings to active replisomes. From a general perspective, our finding that specific functional forms of the MCM complex are 'invisible' to IF, highlights a limitation of studying protein-protein complexes by IF, which may be relevant to many other protein complexes in distinct biological pathways. We provide a solution to this problem by showing that, in the case of the replication machinery, endogenous tagging of MCM subunits enables their detection and quantitative assessment in replication factories (Fig. 4f). Moreover, we explain why previous studies[17,25] using overexpression systems failed to detect GFP-tagged MCMs at active replication sites. We showed that uncontrolled MCM overexpression without removal of endogenous pool of MCMs leads to the production of MCM2-7 complexes which remain largely inactive during DNA replication. This could be caused by overcoming intrinsic MCM maturation trajectories that play an important role in MCM2-7 functioning in successive cell generations[11]. Based on our observations, we emphasize that careful design of protein fluorescent tagging strategies that fully replace endogenous protein levels is critical aspect for the interpretation of imaging data in general.

Conceptually, these data clarify the remaining part of the long-standing MCM paradox[5,10]. The results show that the MCM2-7 complex is the core structural and functional component of active replication forks in cells, fully aligned with the model derived from biochemical and structural studies. Finally, our approach of combining endogenous tagging of MCMs and CMG components (such as CDC45) enables us to unequivocally distinguish a large excess of inactive MCMs from a small fraction of active MCMs that form the core of the replisome (Fig. 4f). The experimental systems and reagents we have developed in the course of this study are amenable to a very broad spectrum of imaging techniques that range from wide-field imaging to super-resolution microscopy. Accordingly, we envisage that direct visualization of the replicative helicase through endogenously tagged components will provide the much-needed impetus for further investigations into replisome dynamics in physiological settings–advancing our understanding of the fundamental mechanisms of DNA replication and the maintenance of genome stability in living eukaryotic cells.

## Methods
### Cell culture
The human osteosarcoma cell line U2OS (ATCC HTB-96) and human immortalized retinal epithelial cell line hTERT-RPE1 (ATCC CRL-4000) authenticated by STR profiling (IdentiCell Molecular Diagnostics) were grown under standard conditions in Dulbecco's modified Eagle's medium (DMEM, high glucose, GlutaMAX supplement, pyruvate; Thermo Fisher Scientific, 31966047) supplemented with 10% fetal bovine serum (FBS, Thermo Fischer Scientific, 10500064) and 0.5% penicillin-streptomycin (Thermo Fischer Scientific, 15140122). CHO cell line ectopically expressing MCM4-GFP, and PCNA-RFP was maintained in DMEM media supplemented with 10% FBS, 0.5 % penicillin-streptomycin, MEM nonessential amino acids solution (Thermo Fischer Scientific, 11140035), 400 μg/ml Zeocin (Thermo Fischer Scientific, R25001), 400 μg/ml G418 (Thermo Fischer Scientific, 10131027), 500 μg/ml Hygromycin B (Thermo Fischer Scientific, 10687010) and 2.5 μg/ml Blasticidin (Thermo Fischer Scientific, A1113903). Expression of MCM4-GFP was regulated by doxycycline (Merck, D9891-5G) concentrations corresponding to the data in the figures. All cell lines and their derivatives were regularly tested for mycoplasma using the MycoAlert detection kit (Lonza, LT07-118). Unless stated otherwise (Supplementary Fig. 6b; Supplementary Fig. 7c–f), all experiments have been performed in U2OS cells or its derivatives with endogenously tagged proteins as specified in Figure legends.

### CRISPR-Cas9 generation of endogenously tagged cell lines
U2OS cells expressing C-terminal endogenously tagged replisome components were generated using CRISPR-Cas9 as described previously[11,28]. Paired guide RNAs (gRNA) for specified genomic locus (POLD1: gRNA 1, AGGGAGAATTAATAAAGTTC and gRNA 2, CGCCCCATGGGATGCTTGCA; POLE1: gRNA 1, TGCCAGGCCTCCT-GATGCCA and gRNA 2, GCAGAGGCACCCGGGGCCCG) were cloned into pX335-U6-Chimeric_BB-CBh-hSpCas9n(D10A) (Addgene plasmid 42335, a gift from F. Zhang) via *Bbs*I restriction site[29]. Naïve U2OS cells were transfected by Lipofectamine LTX Plus reagent (Thermo Fischer Scientific, 15338-100) with pX335 plasmids containing cloned gRNAs and donor plasmid containing GFP tag with flexible linker flanked by 900 bp homology arms complementary to C-terminus of the specific genomic locus. After 10 days, transfected cells were sorted for GFP-positive cells. After cell sorting, cells were serially diluted into 100-mm dishes to obtain single colonies, which were expanded for further characterization and functional validation by western blotting, junction PCR at a specified genomic locus, immunofluorescence (sub-cellular localization), and QIBC. CRISPR-Cas9-mediated generation and validation of CDC45-GFP; CDC45-GFP, MCM4-Halo, and MCM2-GFP U2OS cell lines were described previously[11].

### Chemical reagents, siRNAs, and antibodies
Cells were treated with ATR inhibitor (AZD6738, Selleckchem, S7693) at a final concentration of 5 μM for 1 h in experiments as indicated. siRNA (Ambion Silencer Select) transfection was performed using Lipofectamine RNAiMax (Thermo Fisher Scientific, 13778075) at a concentration of 5 nM siRNA against TIMELESS (s17054), CLASPIN (s34330), and AND1 (s22040) for 24 h. Non-targeting siRNA (Ambion negative control #1) was used as control siRNA.

Primary antibodies for immunofluorescence (IF) were used as follows: GFP (rabbit, Chromotek, PABG1-100, 1:1,000), PCNA (human,

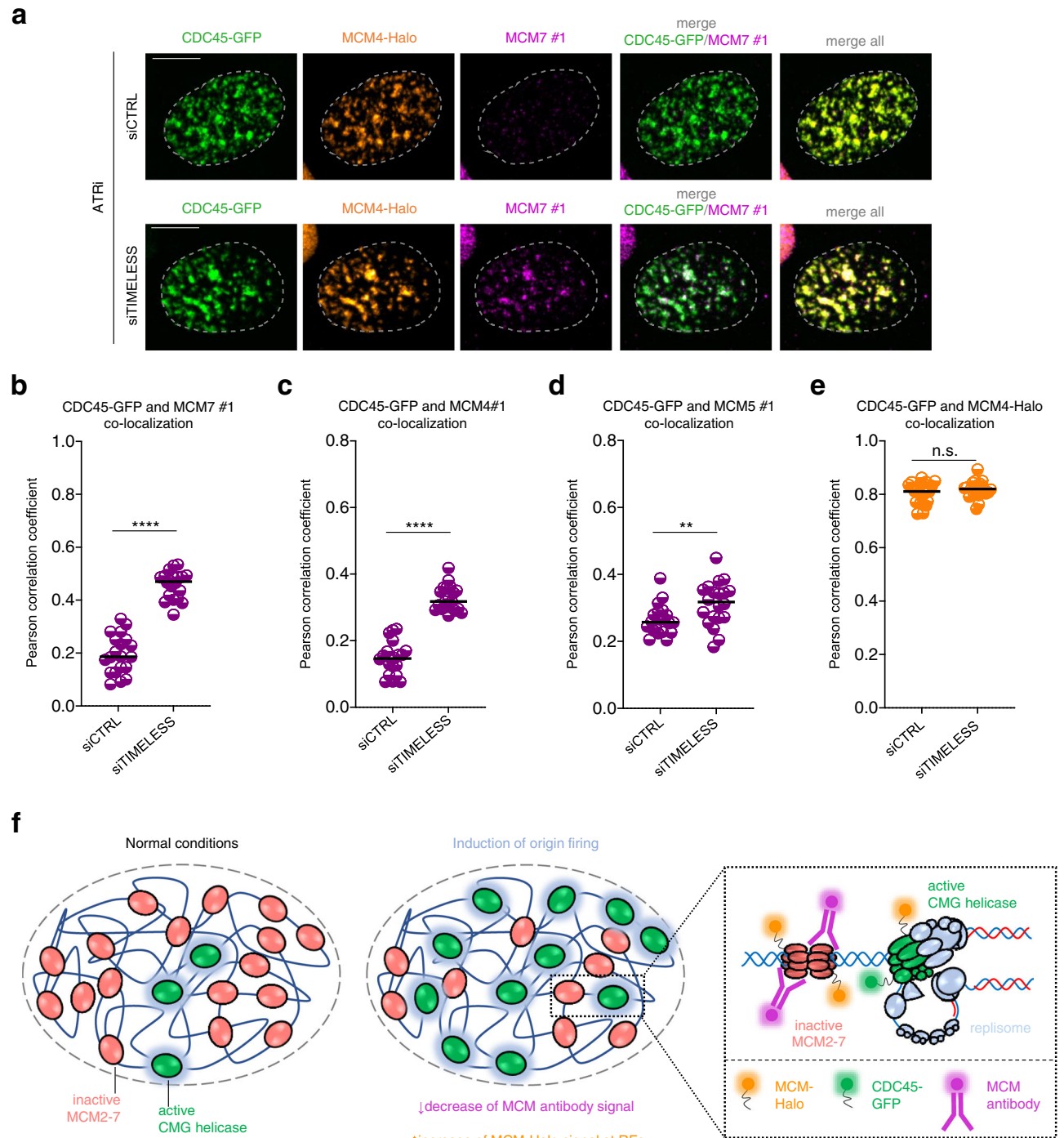

**Fig. 4 | Replisome components limit the MCM accessibility at actively progressing replication forks. a** Representative MIP confocal images of chromatin-bound CDC45-GFP, MCM4-Halo and immunostained MCM7 in S5 stage of S phase sub-stratified as in (Supplementary Fig. 1b). Cells were transfected by control siRNA or siRNA against *TIMELESS* (5 nM; 24 h) as indicated and treated with ATR inhibitor (ATRi). Scale bar, 10 µm. **b** Colocalization analysis of endogenously tagged CDC45-GFP and immunostained MCM7 in the S5 stage of S phase in cells treated as in **a**. Lines are medians; *n* = 20 cells per condition; *P* value was calculated by two-tailed unpaired *t*-test; ****P > 0.0001. **c** Colocalization analysis of endogenously tagged CDC45-GFP and immunostained MCM4 in the S5 stage of S phase in cells treated as in **a**. Lines are medians; *n* = 20 cells per condition; *P* value was calculated by two-tailed unpaired *t*-test; ****P > 0.0001. **d** Colocalization analysis of endogenously tagged CDC45-GFP and immunostained MCM5 in cells

treated as in **a**. Lines are medians; *n* = 20 cells per condition; *P* value was calculated by two-tailed unpaired *t*-test; **P = 0.0096. **e** Colocalization analysis of endogenously tagged CDC45-GFP and MCM4-Halo in the S5 stage of S phase in cells treated as in **a**. Lines are medians; *n* = 20 cells per condition; *P* value was calculated by two-tailed unpaired *t*-test; not significant (n.s.) denotes *P* > 0.05. **f** A model clarifying the MCM paradox. Upon conversion of inactive MCMs to active replisomes, antibody binding sites in the MCM2-7 proteins are masked by replisome factors, hindering MCM immunodetection. Thus, MCM antibodies primarily detect the highly abundant pool of inactive MCMs (in MCM-DHs). In contrast, endogenous tagging of MCM subunits enables visualization of entire pool of MCMs, including those in fully assembled active replisomes (see text for details). Source data are provided as a Source data file.

Immuno Concepts, 2037, 1:1,000), TIMELESS (rabbit, Abcam, ab109512, clone EPR5275, 1:500). Primary antibody for western blotting (WB) were used as follows: AND-1 (rabbit, Abcam, ab224221, 1:1,000), CDC45 (rabbit, Cell Signaling Technology, 11881S, clone D7G6, 1:1,000), CLASPIN (mouse, Santa Cruz, sc-376773, clone B-6, 1:1,000), GFP (rabbit, Chromotek, PABG1-100, 1:1,000), KAP-1 (rabbit, Bethyl Laboratories, A300-274A, 1:2,000), MCM4 #1 (mouse, Novus Biologicals, H00004173-B01P, 1:1000), MCM7 #1 (mouse, Santa Cruz, sc-9966, clone 141.2, 1:1000), PCNA (mouse, Santa Cruz, sc-56, clone PC-10, 1:1,000), POLD1 (rabbit, Abcam, ab186407, clone EPR15118, 1:1,000), POLE1 (rabbit, Abcam, ab226848, 1:1,000), TIMELESS (rabbit, Abcam, ab109512, clone EPR5275, 1:1,000). Secondary antibody conjugates for IF were goat anti-mouse and goat anti-rabbit Alexa Fluor 488 (A11029, A11034), Alexa Fluor 568 (A11031, A11036) Alexa Fluor 647 (A21236, A21245) (all from Thermo Fischer Scientific, 1:1,000) and donkey anti-human Alexa Fluor 647 (Jackson Immuno Research, 709-605-149, 1:1,000). Secondary antibody conjugates for WB were HRP horse anti-mouse IgG antibody (Vector Laboratories, PI-2000, 1:10,000) and HRP goat anti-rabbit IgG antibody (Vector Laboratories, PI-1000, 1:10,000).

## Selection of MCM antibodies

Anti-MCM antibodies recommended for IF staining and with defined antigen sequence were searched among fourteen antibody manufacturers. Representative antibodies for each MCM subunit were selected based on the number of citations according to cite AB online tool[30], location, and length of the antibody antigen, whenever possible, we included antibodies raised against full length protein and both monoclonal and polyclonal antibody types. For MCM2 following antibodies were used: MCM2 #1 (mouse, monoclonal, Novus Biologicals, H00004171-M01, clone 6A8, IF dilution: 1:1,000, 5 citations, antigen location: 805-904 aa), MCM2 #2 (rabbit, polyclonal, Proteintech, 10513-1-AP, IF dilution: 1:1,000, 17 citations, antigen location: 657-904 aa), and MCM2 #3 (mouse, monoclonal, Santa Cruz, sc-373702, clone E-8, IF dilution: 1:1,000, 9 citations, antigen location: 9-34 aa). For MCM3 following antibodies were used: MCM3 #1 (mouse, monoclonal, Santa Cruz, sc-390480, clone E-8, IF dilution: 1:1,000, 5 citations, antigen location: 1-215 aa), MCM3 #2 (rabbit, polyclonal, Antibodies.com, A29136, IF dilution: 1:1,000, not listed in AB cite database, antigen location: full length), and MCM3 #3 (rabbit, polyclonal, Abcam, ab4460, IF dilution: 1:1,000, 24 citations, antigen location: 500-600 aa). For MCM4 following antibodies were used: MCM4 #1 (mouse, polyclonal, Novus Biologicals, H00004173-B01P, IF dilution: 1:1,000, 0 citations, antigen location: full length), and MCM4 #2 (mouse, monoclonal, Santa Cruz, sc-28317, clone G-7, IF dilution: 1:1,000, 17 citations, antigen location: 1-300 aa). MCM5 #1 (rabbit, polyclonal, Abcam, ab17967, IF dilution: 1:1,000, 24 citations, antigen location: 1-55 aa) antibody was used to detect MCM5 subunit. For MCM6 following antibodies were used: MCM6 #1 (rabbit, monoclonal, Abcam, ab201683, clone EPR17686, IF dilution: 1:1,000, 12 citations, antigen location: 650-821 aa), and MCM6 #2 (mouse, monoclonal, Novus Biologicals, H00004175-M04, clone 7D8, IF dilution: 1:1,000, 0 citations, antigen location: full length). MCM7 #1 (mouse, monoclonal, Santa Cruz, sc-9966, clone 141.2, IF dilution: 1:1,000, 157 citations, antigen location: full length) was used to detect MCM7 subunit. Among all MCM antibodies, the MCM7 antibody has the highest citation rate and therefore was used as a reference MCM antibody in this manuscript.

## Western blotting

Whole cell lysates were obtained by cell lysis in buffer (10 mM Hepes pH 7.5, 500 mM NaCl, 1 mM EDTA, 1% NP-40) supplemented with protease and phosphatase inhibitors (ROCHE, 04693116001 and 04906837001) and 750 U per ml of benzonase (Sigma-Aldrich, E1014-25KU). Revert Total Protein Stain (Licor, 926-11021) was used for total

protein staining following the manufacturer's recommendations. KAP-1 and PCNA antibodies were used as loading controls. Loading controls (Supplementary Figs. 1a, d and 9a) were processed from the same membrane used to detect the target proteins after stripping the membrane with Restore PLUS Western Blot Stripping Buffer (Thermo Fischer Scientific, 46430). Processing controls (Supplementary Fig. 1e) were run in parallel using the same lysates and identical loading concentrations were used to detect the target proteins. Uncropped scans of all western blots and gels are provided in Source data file.

## IF staining

Cells were grown on round 12-mm diameter, 1.5-mm-thick glass coverslips (cleaned in 96% ethanol, dried, and autoclaved). For immunostaining of chromatin-bound proteins, cells were pre-extracted with ice-cold PBS containing 0.2% TritonX-100 (Sigma-Aldrich, T9284-500ML) for 2 min on ice before fixation by 4% buffered formaldehyde (VWR Chemicals, 9713.1000) for 15 min at room temperature. Pre-extraction protocol applying increased stringency of detergent used in Supplementary Fig. 3c, d was performed with ice-cold 0.5% TritonX-100 (Sigma-Aldrich, T9284-500ML) in CSK buffer (10 mM Hepes pH 7.5, 300 mM Sucrose, 100 mM NaCl, 3 mM MgCl$_2$) for 10 min on room temperature followed by fixation with 4% buffered formaldehyde (VWR Chemicals, 9713.1000) for 30 min at room temperature.

For HaloTag labeling, total pool of chromatin-bound MCM4-Halo was detected by HaloTag ligand as described previously[11]. In brief, cells expressing endogenously tagged MCM4-Halo were pulsed with JF549 HaloTag ligand (Promega, GA1111) at a final concentration of 200 nM for 20 min before pre-extraction and fixation. For Click-iT EdU staining, cells were incubated with 10 µM EdU (Thermo Fisher Scientific, 31985070) for 15 min before pre-extraction and fixation. EdU detection was performed according to the manufacturer's recommendations before incubation with primary antibodies.

Primary and secondary antibodies were diluted in DMEM media containing 10% FBS and 0.05% sodium azide (filtered through a 0.2 µm filter) and incubated at room temperature for 90 min and 45 min, respectively. Immunostaining using MCM6 #1 antibody was performed as described[31]. Briefly, after pre-extraction and fixation, coverslips were blocked in PBS buffer containing 2% glycine, 2% BSA, 0.2% gelatin, and 50 mM NH$_4$Cl overnight before IF staining. For DAPI staining, a secondary antibody solution was supplemented with 0.5 µg ml$^{-1}$ 4′,6′-diamidino-2-phenylindole-dihydrochloride (DAPI, Thermo Fischer Scientific, D1306). After staining, coverslips were washed three times with PBS and additionally twice in distilled water, dried, and mounted with a Mowiol-based mounting medium (12% Mowiol 4-88 (Sigma-Aldrich, 81381), 30% glycerol, 0.12 M Tris-HCl pH 8.5).

## Quantitative image-based cytometry

Images were acquired using ScanR inverted high-content screening microscope (Olympus) equipped with wide-field optics, UPLSAPO dry objective (20×, 0.75-NA), fast excitation and emission filter-wheel devices for DAPI, FITC, Cy3, and Cy5 wavelengths, an MT20 illumination system, and a digital monochrome Hamamatsu ORCA-R2 CCD camera. Images were acquired in an automated fashion with the ScanR acquisition software (Olympus, v.2.7.1) at non-saturating conditions (12-bit dynamic range) by maintaining constant laser intensity at 100% and adjusting the exposure times for individual fluorophores. Detailed acquisition information for each experiment can be found in the source data for QIBC. At least 5000 cells per condition were acquired. Acquired images were processed and analyzed with ScanR analysis software (Olympus, v.2.7.1). An automated dynamic background correction (thresholding at least five-fold pixel intensity above background levels) was applied to all images for each channel separately but maintaining similar parameters for all treatments within an experiment. The DAPI signal was used for the generation of an intensity-threshold-based mask to

identify individual nuclei as main objects. This mask was then applied to analyze pixel intensities in different channels for each nucleus. A table with values was then exported and analyzed in Spotfire software (Tibco, v.10.5.0.72). Within one experiment, similar cell numbers were compared for the different conditions. Color-coded scatter diagrams were converted to Excel (Microsoft, v.16.57) spreadsheets and are provided in Source data.

## Confocal microscopy

Confocal images were acquired using an UltraVIEW Vox spinning-disk microscope (Perkin Elmer) equipped with a 60×, 1.4-NA Plan-Apochromat oil immersion objective, excitation, and emission filter-wheel devices for DAPI, FITC, Cy3, and Cy5 wavelengths, and a Hamamatsu EMCCD 16-bit camera. Images were acquired using Velocity software (v.6.3), in which the laser power and exposure times were appropriately adjusted to avoid saturated intensities and were maintained identical within a series of experiments. Microscope performance and channel alignment were regularly checked by imaging 200-nm multicolor fluorescent beads. For optimal representations in figures, images were adjusted for brightness and contrast and were exported as 16-bit TIFF files. For colocalization analysis, a thresholded Pearson correlation coefficient was calculated by the Volocity Quantitation Colocalization feature within Volocity software (v.6.3) as described[32]. The DAPI signal was used to identify individual nuclei as main objects, in which only intensity values over a determined threshold separating signal from a background in each channel were included in the Pearson correlation coefficient calculation. Determined thresholds for each channel were applied to all images maintaining similar parameters for all treatments within an experiment. Similar cell numbers were compared for the different conditions within one experiment.

## Statistical analysis

Statistical analysis was performed using a two-tailed unpaired *t*-test in GraphPad Prism 9.0.0. Experiments were not randomized, and no blinding was used during data analysis. Sample size, statistical tests, and the number of replicates for all experiments are specified in the figure legends.

## Reporting summary

Further information on research design is available in the Nature Research Reporting Summary linked to this article.

## Data availability

Primary imaging data have been deposited at the European Bioinformatics Institute (EBI) BioStudies database under accession code S-BSST909. The electron microscopy data for *S. cerevisiae* MCM2-7 double hexamers used in this study are available in the Protein Data Bank database under accession code 6F0L. The electron microscopy data for human core leading strand replisome used in this study are available in the Protein Data Bank database under accession code 7PFO. Any additional data or information in support of this study is available from corresponding authors upon reasonable request. Source data are provided with this paper.

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

## Acknowledgements

Research funding was provided by the Novo Nordisk Foundation (NNF14CC0001). H.P.-S. was supported by the Czech Science Foundation Junior Star (22-20303M). T.C.R.M. was supported by the Danish National Research Foundation (DNRF115) and Carlsberg Foundation (CF21-0571). We thank Protein Imaging Platform at the Center for Protein Research for the technical assistance. Cell sorting was carried out at CPR and Danstem Flow Cytometry Platform. We thank D. Gilbert for sharing the CHO cell line ectopically expressing MCM4-mEmerald and PCNA-RFP. The pX335 plasmid was a gift from F. Zhang. We thank all members of the Lukas and Sedlackova lab for stimulating discussions and critical comments on the manuscript.

## Author contributions

H.P.-S. and J.L. conceived the project. H.P.-S. devised the original concept, designed, and performed experiments, analyzed the data, and prepared figures. T.C.R.M. carried out the structural analysis. H.P.-S. generated all the cell lines with the help of M-B.R. and J.K. H.P.-S. and J.L. wrote the manuscript. All authors read and commented on the manuscript.

## Competing interests

The authors declare no competing interests.
