## [Peer Review File · Nature Communications]

Solving the MCM paradox by visualizing the scaffold of CMG helicase at active replisomesREVIEWER COMMENTS

Reviewer #1 (Remarks to the Author):

In this manuscript, the authors have resolved a longstanding paradox in the replication field, that is, by all accounts MCM is the replicative helicase but it does not co-localize with replication forks in living cells. Many hypotheses have been proposed to explain this "MCM paradox". Here the authors test the hypothesis that epitope masking by the myriad proteins that interact with the MCM active but not dormant complex prevents its detection. They show that detecting MCM by fusion to a fluorescent tag (Halo) gives results that are fully consistent with a replication fork protein that is also bound to chromatin prior to replication. The paper is elegant in its simplicity and the results certainly are convincing and of interest to people in the replication field. They also provide another example of epitope masking for cell biologists in general as the authors point out. My major concern is that the work may not be fully accepted by the field because it does not satisfactorily explain why this has not been observed before. MCM has been fluorescently tagged by many groups but, to my knowledge, the fluorescent MCM still does not co-localize with DNA synthesis or with replication fork proteins. The authors are so careful to address other aspects of the literature it is a glaring omission, particularly as there is plenty of space to develop it. Best would be for them to look at those papers carefully, propose a hypothesis and test that hypothesis. Then this story would be complete and convincing for the community.

Another specific point relates to the TIMELESS knockdown. There are several issues of concern here. First, TIMELESS depletion leads to major replication fork problems and DNA damage. How was this taken into account. Secondly, and very importantly, what is the rationale for the focus on TIMELESS? It doesn't seem to mask any epitopes in the structure (Fig. 1e). The major problem here would be if the rationale is that it gives the result they are looking for. For this experiment to be convincing, we should see knockdown of several proteins, and the increase in signal resulting from those knockdowns should be related to the epitope that they mask sterically. That experiment should include negative control knockdowns of proteins that do not mask and do not alter the immunofluorescence signal.

Minor point- In EDF 1 d, GFP is mislabeled.

Reviewer #2 (Remarks to the Author):

The Mcm2-7 catalytic core of the eukaryotic CMG replicative helicase has never been visualised at sites of active DNA replication in mammalian cells. Numerous hypotheses have been put forward for this apparent MCM 'paradox', but experimental evidence for any of these explanations is lacking. In the current manuscript, the authors seek to address this issue, using a range of microscopic imaging techniques in mammalian cells. In microscopy experiments, the authors convincingly demonstrate that antibodies raised against endogenous Mcm2-7 proteins detect Mcm2-7 double hexamers but do not readily recognise Mcm2-7 as part of the CMG complex at replication forks, most likely due to steric occlusion of Mcm2-7 epitopes by accessory replisome factors such as TIMELESS. This problem can be circumvented by tagging individual Mcm2-7 subunits (e.g. Mcm4-Halo), which allows visualisation of Mcm2-7 at sites of replication, as evidenced by co-localisation with Cdc45, PCNA and EdU foci.

The major advance of the paper seems to be technical, and I do not doubt that the experimental approach described will be very useful for the DNA replication field, and enable future studies on replisome (particularly Mcm2-7) dynamics in mammalian cells. Furthermore, the experimental data are of high quality and clearly presented, and the manuscript is focussed and logical. However, as the authors acknowledge, that Mcm2-7 is part of the CMG and replisome in human cells is no great surprise, given the huge wealth of data in support of this notion across multiple eukaryotic species,

and the recent cryo-EM structures of the human replisome with Mcm2-7 at its core.

Specific points:

- The failure of the anti-MCM antibodies (detailed in the Extended Data table) to detect CMG is a key feature of this paper. Are these the standard, accepted anti-MCM antibodies used by the field, or are there others? If there are other antibodies available, how did the authors select with antibodies to use?
- It would be useful for the reader if the authors could highlight the position of the Mcm4-Halo and Mcm2-GFP tags on a replisome structure, analogous to the depictions in Figure 2e, f
- It would be nice if the authors could be more systematic in their approach (also relevant to the above point about antibody selection). For example, does siRNA against other replisome proteins (e.g. CLASPIN, AND-1) allow detection of Mcm2-7 at replication forks, or did the authors only try TIMELESS depletion? Also, did the authors try tagging other Mcm2-7 subunits in addition to Mcm2 and 4? Given the technical nature of the paper, this type of information would be very useful.
- It seems pretty clear that the differences in the ability of the anti-MCM and anti-tag antibodies to detect Mcm2-7 as part of CMG is not due to intrinsic differences in the sensitivities of the various antibodies (for example, the TIMELESS data in Figure 4 argues against this). This fact should be stated for clarity.
- The fact that TIMELESS depletion enhances the co-localised signal between endogenous Mcm2-7 and Cdc45 but not HALO-tagged Mcm4 and Cdc45 is an informative control, as it shows that the TIMELESS effect is most likely due to steric occlusion of Mcm2-7 epitopes rather than, for example, any impact upon DNA replication per se, or replisome disassembly (TIMELESS is required for optimal replisome disassembly in worms – Xia et al, EMBO J, 2021). Again, a statement on this point would be helpful for clarity.

Reviewer #3 (Remarks to the Author):

In the manuscript: "Solving the MCM paradox by visualizing the MCM scaffold of endogenous CMG helicase at active replisomes" by Dr Polasek-Sedlackova and others, the authors revisit the so called MCM paradox i.e the inability to detect MCMs at active replication forks by immunofluorescence despite it having an essential function in replication elongation. In part the reason behind the paradox is existence of many folds higher number of inactive MCM2-7 dormant origins that are positioned in yet unreplicated parts of the genome and can mask the detection of MCM2-7 within active replisomes. Through population based quantitative immunofluorescence (QIBC) and CRISPR aided endogenous tagging of MCMs, the authors concluded that the steric hindrance caused during conversion of MCM-DHs to active CMGs leads to epitope masking thereby causing limited detection by antibody-based immunofluorescence (IF). Despite having done very decisive and clear experiments to prove their claim, the manuscript seems more suitable as being titled as a methodological development in detection of MCMs at active replication forks rather than reporting any novel biological phenomenon/activity. In summary, it is a nice study with robust experiments, but it does seem to bring very limited novel biological discovery.

Major points

1. Authors set up the scene as if there were doubts whether MCM2-7 form part of a replicative helicase ("Are MCM proteins parts of functional replisomes in cells?", line 52). However, in my mind there is no such doubt in the literature at all, as many studies isolated MCM2-7 as part of active

replisomes or shown their role in DNA unwinding in different ways (e.g. Labib et al.,2000; Calzada et al.,2005; Gambus et al.,2006; Pacek et al.,2006; Mašata et al.,2011; Labib et al.,2017).

2. Authors conclude that limited detection of MCM2-7 within active replisomes by antibodies is due to the fact that their epitopes are masked by other replisome components and so these antibodies are much better at detecting inactive MCM2-7 than the ones within the large replisome machinery. It would be however nice if more than one antibody was tested for each of the MCM subunits. In fact, most of the antibodies used in this study (except for MCM5) are monoclonal antibodies or polyclonal antibodies against a short synthetic peptide. In each case the antibody detects only a very small epitope within the MCM protein and so indeed can be affected by interacting partners of this protein. It would be ideal to test a catalogue of polyclonal antibodies raised against large chunks of the antigen to really prove that MCMs within replisomes are undetectable by antibody based IF.

3. Others also used tagged MCM2-7 (e.g. Symeonidou et al. 2013) and come to a different conclusion – their failed to detect a clear interaction of tagged MCM2-7 with sites of replication. Why the discrepancy?

4. The authors show that detection of endogenously tagged MCMs or CDC45 can overcome the problems of using epitope specific antibodies and detect MCM2-7 within the active replisome. However, protein tagging, especially with large size GFP tag, comes with its own problems – as authors point out themselves these are tight protein complexes that are changing dynamically during the cell cycle and large tags on some of their components can change complex dynamics and formation. I think it is essential that authors verify cell proliferation and complex interactions in cells with tagged MCMs and CDC45.

5. Replisome disassembly is prevented by the presence of the lagging strand during replication elongation and not by masking of the replisome subunits during replication termination. Therefore, preventing replisome disassembly may not be the best example of the physiological relevance of the masking of MCMs caused during conversion to active CMGs.

Minor points:

1. In Fig. 3F middle panel, the Pearson correlation coefficient at S1 and S2 do not vary significantly, contradicting their representative images in Fig. 3E. A similar problem exists with EDF. 6A and B.

2. Page 4: line 123: "increased retention" should be rephrased as "increased detection".

3. The table listing epitope sequence of antibodies should be renamed as antigen sequence.

4. The range of CDC45-GFP signals in unperturbed conditions (DMSO) in Fig. 2A is very low despite being identical to conditions in Fig 1C, and EDF. 1B. In fact, the increased signals upon ATR inhibition seem more like the unperturbed ones in Fig 1C, and EDF. 1B.

5. It is not clear why in EDF. 4C and Fig 3: A-F a particular endogenous MCM subunit is compared to the tagged version of a different MCM subunit (MCM7 vs MCM4-Halo/MCM3 vs MCM2-GFP).

6. Page 3: line 97-99: If the MCM5 epitope is involved in MCM2-7 dimerization, it is expected to be masked from antibody recognition in double hexamers as much as in CMG. It is therefore not expected to be detected by IF both in MCM-DHs as well as active CMG configurations.

7. While the authors claim the use of CDC45 at active RFs as a proxy to distinguish between the bulk of MCMs from the active ones at the RFs as a novel approach, colocalization with PCNA and EdU are equally good proxy indicators used in several studies in the past.

Calzada et al., Molecular anatomy and regulation of a stable replisome at a paused eukaryotic DNA replication fork. *Genes Dev* (2005)

Gambus et al., GINS maintains association of Cdc45 with MCM in replisome progression complexes at eukaryotic DNA replication forks. *Nat Cell Biol* (2006)

Labib et al., Uninterrupted MCM2-7 Function Required for DNA Replication Fork Progression. *Science* (2000).

Labib et al., MCM2-7 Proteins Are Essential Components of Prereplicative Complexes that Accumulate Cooperatively in the Nucleus during G1-phase and Are Required to Establish, But Not Maintain, the S-phase Checkpoint. *Molecular Biology of the Cell* (2017)

Mašata et al., A fraction of MCM 2 proteins remain associated with replication foci during a major part of S phase. *Folia Biologica (Praha)* (2011)

Pacek et al., Localization of MCM2-7, Cdc45, and GINS to the site of DNA unwinding during eukaryotic DNA replication. *Mol. Cell* (2006)

Symeonidou et al., Multi-step Loading of Human Minichromosome Maintenance Proteins in Live Human Cells. *Journal of Biological Chemistry* (2013)

POINT-BY-POINT RESPONSE TO THE REVIEWER'S COMMENTS

We would like to thank all three reviewers for their enthusiasm, valuable comments, and excellent guidance on how to strengthen our manuscript. We found their suggestions very insightful, and we have ensured that all concerns are now addressed in the revised manuscript. Specifically, we want to highlight three major additions that in our view significantly strengthened our original conclusions:

- To validate that immunostained MCMs are primarily detecting inactive MCMs, we now reproduced all previous findings and conclusions with a much larger panel of representative MCM antibodies that were selected based on the number of citations, length, and location of antigen, and antibody type.
- To provide additional evidence that replisome components are sterically shielding active MCMs at a fully assembled replication fork, we depleted all three subunits (TIMELESS, CLASPIN, AND-1) of the replication protection complex (RPC). To our delight, these experiments fully confirmed the prediction from a structural model where the knockdown of the RPC components that directly bind the MCM scaffold of CMG helicase (TIMELESS and CLASPIN) unmasked MCM antibody binding sites, while depletion of AND-1, which interacts with MCM scaffold indirectly via CDC45 and GINS, did not. In our view, the addition of the AND-1 specificity control is really important, and we are grateful to the reviewers for inspiring us to elaborate on this part of our manuscript.
- Finally, we provide an explanation, of why previous studies using ectopically overexpressing GFP-tagged MCMs failed to detect active MCMs in replication sites by showing that overexpressed MCMs preferentially form inactive (dormant) replication origins.

We are very happy about all these amendments because they strengthened the key conclusions, namely that various components of active replisomes shield the MCM scaffold and therefore limit the accessibility of MCM antibodies. We trust that the CRISPR-Cas9 genome editing approach that bypasses the steric occlusion of MCM subunits will provide a major and positive boost in elucidating DNA replication dynamics in a truly physiological context (nucleus of a living mammalian cell). All these new additions and amendments are detailed below, point-by-point.

Reviewer #1 (Remarks to the Author):

In this manuscript, the authors have resolved a longstanding paradox in the replication field, that is, by all accounts MCM is the replicative helicase but it does not co-localize with replication forks in living cells. Many hypotheses have been proposed to explain this "MCM paradox". Here the authors test the hypothesis that epitope masking by the myriad proteins that interact with the MCM active but not dormant complex prevents its detection. They show that detecting MCM by fusion to a fluorescent tag (Halo) gives results that are fully consistent with a replication fork protein that is also bound to chromatin prior to replication. The paper is elegant in its simplicity and the results certainly are convincing and of interest to people in the replication field. They also provide another example of epitope masking for cell biologists in general as the authors point out.

We are very grateful to the Reviewer for his/her appreciation for our work and for the following helpful suggestions to improve our manuscript.

My major concern is that the work may not be fully accepted by the field because it does not satisfactorily explain why this has not been observed before. MCM has been fluorescently tagged by many groups but, to my knowledge, the fluorescent MCM still does not co-localize with DNA

synthesis or with replication fork proteins. The authors are so careful to address other aspects of the literature it is a glaring omission, particularly as there is plenty of space to develop it. Best would be for them to look at those papers carefully, propose a hypothesis and test that hypothesis. Then this story would be complete and convincing for the community.

This is a very important comment raised by the Reviewer. To address this puzzling discrepancy, we obtained CHO cells (from *Kuipers et al, JCB, 2011*) overexpressing MCM4-mEmerald under tet-off promoter and PCNA-RFP as a marker of replication factories. First, we titrated levels of doxycycline to monitor MCM4-mEmerald expression. As described previously upon highest expression of MCM4-mEmerald (0 ng/ul of doxycycline), the level of endogenous untagged MCM4 is slightly decreased, however large fraction is still present (Supplementary Fig. 7c). In these conditions, we performed QIBC analysis (Supplementary Fig. 7d) which showed similar kinetics of MCM4-mEmerald on chromatin as observed previously observed for MCMs visualized by immunostaining or CRISPR-Cas9 endogenous tagging (Supplementary Fig. 2, Fig. 3a, Supplementary Fig. 5d). Next, we performed a colocalization analysis which to our surprise revealed a decrease in the Pearson correlation coefficient between MCM4-mEmerald and PCNA-RFP-labeled RFs after ATR inhibitor treatment (Supplementary Fig. 7e) – very similar to what we report for immunostained MCMs in general (Fig. 2f, Supplementary Fig. 5b). Such a striking observation indicated that overexpressed MCMs become largely engaged as inactive (dormant) replication origins. One reason for this could be that ectopic MCMs may overcome intrinsic regulatory pathways important for proper MCM2-7 functioning during the DNA replication program. In our previous work (*Sedlackova et al, Nature, 2020*), we showed that cellular MCM equilibrium relies on a delicate balance between recycling and biogenesis pathways giving a rise to two different protein pools, parental and nascent MCMs. While parental MCMs are preferentially converted to CMG helicases (even upon increased origin firing by ATR inhibitor), nascent MCMs produced in a specific ratio remain largely inactive during DNA replication in daughter cells. Our work clearly showed that such tight regulation of MCM equilibrium is essential for error-free DNA replication. Therefore, to understand whether MCM overexpression disrupts such equilibrium, we tested their ability to form CMGs by immunoprecipitation analysis. Aligned with our hypothesis, we observed that MCM4-mEmerald possesses a very low ability to interact with CDC45 in direct comparison to endogenously tagged MCM4 (Supplementary Fig. 7f). This indicates that ectopically expressed MCMs behave largely as inactive replication origins. We also note that previous papers attempted to ectopically express MCMs close to physiological level (*Kuipers et al, JCB, 2011, Symeonidou et al, JBC, 2013*), but in all such cases untagged endogenous MCMs were not removed from cells, inevitable leading to an imbalance in parental versus nascent MCM pools and thus precluding to visualize active MCMs by immunostaining. Moreover, our data point out that careful design of protein fluorescent tagging (such as CRISPR-Cas9 endogenous tagging) leading to complete replacement of untagged endogenous protein levels, which is thoroughly validated by multiple approaches, is a critical aspect for interpretation of imaging data in general.

Another specific point relates to the TIMELESS knockdown. There are several issues of concern here. First, TIMELESS depletion leads to major replication fork problems and DNA damage. How was this taken into account. Secondly, and very importantly, what is the rationale for the focus on TIMELESS? It doesn't seem to mask any epitopes in the structure (Fig. 1e). The major problem here would be if the rationale is that it gives the result they are looking for. For this experiment to be convincing, we should see knockdown of several proteins, and the increase in signal resulting from those knockdowns should be related to the epitope that they mask sterically. That experiment should include negative control knockdowns of proteins that do not mask and do not alter the immunofluorescence signal.

We agree with the Reviewer that the rationale behind TIMELESS depletion was not adequately described in a previous version of our manuscript, and we made a major effort to clarify and extend this part of the manuscript. To expand evidence that active MCMs are hidden by replisome factors at fully assembled replication forks, we now extended the TIMELESS data by

knocking down also the two additional subunits of the replication protection complex (RPC), namely CLASPIN and AND-1. As we detail below, the RPC complex is uniquely suitable for addressing this question because the currently available structural knowledge (Fig. 2h) enabled us to predict which RPC component should, or should not, unmask the MCM antibody binding sites. Indeed, the structural analysis revealed that TIMELESS is shielding MCM7, MCM4, MCM6, and MCM2 subunits. CLASPIN interacts with CMG at multiple locations, particularly with MCM6 and MCM2. Importantly, AND-1 is linked to the MCM scaffold only indirectly via CDC45 and GINS thereby representing an interesting negative control for our knockdown experiments as suggested by the Reviewer. Furthermore, as all RPC subunits are accessory replisome components, it was possible for us to find experimental conditions, which do not cause extensive changes in the core replisome architecture leading to replication fork stalling and massive DNA damage. In our previous work (Somyajit *et al*, *Science*, 2017), we showed that siRNA-mediated depletion of TIMELESS, CLASPIN, AND-1 for short time (24 hours) slows down (but does not stop) DNA replication, which is the time frame of our experiments causing only negligible DNA damage. With these assertions and tools in hand, we generated and included in the revised manuscript a set of new results, which fully support our model and the essence of which is summarized in the following section of this rebuttal.

Aligned with our structural predictions, TIMELESS depletion unmasked MCM7 and MCM4 subunits leading to an increase in the Pearson correlation coefficient (Fig. 4a-c). Only moderate effect was observed for the MCM5 subunit (Fig. 4d) due to its opposite location to the identified TIMELESS-CMG interaction interface (Fig. 2h). Similarly, CLASPIN knock-down lead to increased detection of CDC45-marked RFs by MCM2 and MCM6 antibodies (Supplementary Fig. 9b-d), while colocalization between CDC45 and MCM7 subunit located opposite the CLASPIN binding site remained at low levels (Supplementary Fig. 9e). On the other hand, AND-1 depletion had no effect on visualization of RFs by various MCM antibodies as the AND-1 interaction with CMG is mediated primarily through CDC45 and GINS and not MCMs (Supplementary Fig. 9g-j). Importantly, the antibody-independent co-localization of endogenously tagged MCM4-Halo with CDC45 was unaffected by RPC depletions (Fig. 4a, e, Supplementary Fig. 9b, f, g, k) confirming the specificity of the above results and excluding major adverse effects on DNA replication per se.

Minor point- In EDF 1 d, GFP is mislabeled.

We have now corrected the mislabeling in EDF1d (currently Supplementary Fig. 1d).

Reviewer #2 (Remarks to the Author):

The Mcm2-7 catalytic core of the eukaryotic CMG replicative helicase has never been visualised at sites of active DNA replication in mammalian cells. Numerous hypotheses have been put forward for this apparent MCM 'paradox', but experimental evidence for any of these explanations is lacking. In the current manuscript, the authors seek to address this issue, using a range of microscopic imaging techniques in mammalian cells. In microscopy experiments, the authors convincingly demonstrate that antibodies raised against endogenous Mcm2-7 proteins detect Mcm2-7 double hexamers but do not readily recognise Mcm2-7 as part of the CMG complex at replication forks, most likely due to steric occlusion of Mcm2-7 epitopes by accessory replisome factors such as TIMELESS. This problem can be circumvented by tagging individual Mcm2-7 subunits (e.g. Mcm4-Halo), which allows visualisation of Mcm2-7 at sites of replication, as evidenced by co-localisation with Cdc45, PCNA and EdU foci.

The major advance of the paper seems to be technical, and I do not doubt that the experimental approach described will be very useful for the DNA replication field and enable future studies on replisome (particularly Mcm2-7) dynamics in mammalian cells. Furthermore, the experimental data

are of high quality and clearly presented, and the manuscript is focused and logical. However, as the authors acknowledge, that Mcm2-7 is part of the CMG and replisome in human cells is no great surprise, given the huge wealth of data in support of this notion across multiple eukaryotic species, and the recent cryo-EM structures of the human replisome with Mcm2-7 at its core.

We thank the Reviewer for appreciating our efforts in establishing new tools and ways how to explore replisome dynamics *in vivo*. We agree that recent biochemical and structural studies reconstituting the replisome with purified human replication fork factors clearly demonstrate that MCM2-7 is part of the CMG and replisome. However, the inability to detect MCMs at active replication sites in the cellular environment led to several doubts including those considering alternative replisome architecture in the cellular environment. As our data clearly show, the explanation of this “MCM paradox” is ultimately very straightforward — commonly used MCM antibodies cannot access the MCM scaffold shielded by various replisome factors at fully assembled replication forks. Although simple, we believe that our work provides important validation of replisome architecture in physiological settings and at the same time opens new avenues for exciting discoveries of replisome dynamics directly in living cells.

Specific points:

- The failure of the anti-MCM antibodies (detailed in the Supplementary table) to detect CMG is a key feature of this paper. Are these the standard, accepted anti-MCM antibodies used by the field, or are there others? If there are other antibodies available, how did the authors select with antibodies to use?

This is an important point raised by the Reviewer. We have now basically doubled the amount of MCM antibodies used in the original manuscript and for all of them we now provide a detailed description in the method section on how they were selected. Briefly, we searched for MCM antibodies suitable for IF staining among fourteen companies (Abcam, AB clonal, Antibodies.com, Atlas Antibodies, Bio-Rad Laboratories, Bioss, Cell Signaling Technology, Invitrogen, LSBio, Novus Biologicals, Proteintech, RayBiotech, Santa Cruz and St John's Laboratory). Representative antibodies for each MCM subunit were selected based on the number of citations according to the cite AB online tool, location, and length of the antibody antigen. Whenever possible, we included antibodies raised against full length protein and both monoclonal and polyclonal antibody types. As extended information for the reviewers, a detailed description, and graphical summary of antibody selection for each MCM subunit are summarized below.

For MCM2 (**Rebuttal Fig. 1**), 48 antibodies suitable for IF staining were found (21 peptide/fragment antibodies with defined sequence, 21 peptide/fragment antibodies with undefined sequence, 0 antibodies against full length protein sequence, 6 undefined antibodies (proprietary to its manufacturer)). All undefined antibodies were excluded from further selection. **From 21 antibodies with a defined sequence, we selected 3 representatives: #1 (Novus Biologicals, H000041-M01), #2 (Proteintech, 10513-1-AP), and #3 (Santa Cruz, sc-373702).**

Rebuttal Fig. 1. A graphical summary of MCM2 antibody selection. a) Definition of antigen sequence in MCM2 collection of antibodies suitable for IF. **b)** A table of MCM2 antibodies with defined sequence including information on host, clonality and number of citations according to AB cite online tool. **c)** Antigen localization of MCM2 antibodies listed in (b).

For MCM3 (Rebuttal Fig. 2), 34 antibodies suitable for IF staining were found (23 peptide/fragment antibodies with defined sequence, 8 peptide/fragment antibodies with undefined sequence, 1 antibody against full length protein sequence, 2 undefined antibodies (proprietary to its manufacturer)). All undefined antibodies were excluded from further selection. **From 24 antibodies with a defined sequence, we selected 3 representatives: #1 (Santa Cruz, sc-390480), #2 (Antibodies.com, A29136), and #3 (Abcam, ab4460).**

Rebuttal Fig. 2. A graphical summary of MCM3 antibody selection. a) Definition of antigen sequence in MCM3 collection of antibodies suitable for IF. b) A table of MCM3 antibodies with defined sequence including information on host, clonality and number of citations according to AB cite online tool. c) Antigen localization of MCM3 antibodies listed in (b).

For MCM4 (Rebuttal Fig. 3), 19 antibodies suitable for IF staining were found (14 peptide/fragment antibodies with defined sequence, 2 peptide/fragment antibodies with undefined sequence, 2 antibodies against full length protein sequence, 1 undefined antibody (proprietary to its manufacturer)). All undefined antibodies were excluded from further selection. **From 16 antibodies with a defined sequence, we selected 2 representatives: #1 (Novus Biologicals, H00004173-B01P), and #2 (Santa Cruz, sc-28317).** We also tested #3 (Abcam, ab4459), however, this antibody did not pass QIBC validation (Rebuttal Fig. 3d).

Rebuttal Fig. 3. A graphical summary of MCM4 antibody selection. a) Definition of antigen sequence in MCM4 collection of antibodies suitable for IF. b) A table of MCM4 antibodies with defined sequence including information on host, clonality and number of citations according to AB cite online tool. c) Antigen localization of MCM4 antibodies listed in (b). d) QIBC of CDC45-GFP cells immunostained for chromatin-bound PCNA and MCM4 #3. Nuclear DNA was counterstained by DAPI. PCNA was used as cell cycle marker; $n \approx 10,000$ cells per condition. A.U., arbitrary units.

For MCM5 (Rebuttal Fig. 4), 42 antibodies suitable for IF staining were found (28 peptide/fragment antibodies with defined sequence, 8 peptide/fragment antibodies with undefined sequence, 3 antibodies against full length protein sequence, 3 undefined antibodies (proprietary to its manufacturer)). All undefined antibodies were excluded from further selection. **From 31 antibodies with a defined sequence, we selected one representative: #1 (Abcam, ab17967).** We tried to obtain the full length MCM5 antibody but even after several months we failed to obtain these antibodies from two different companies therefore we used only one representative MCM5 antibody in our study.

Rebuttal Fig. 4. A graphical summary of MCM5 antibody selection. a) Definition of antigen sequence in MCM5 collection of antibodies suitable for IF. **b)** A table of MCM5 antibodies with defined sequence including information on host, clonality and number of citations according to AB cite online tool. **c)** Antigen localization of MCM5 antibodies listed in (b).

For MCM6 (Rebuttal Fig. 5), 28 antibodies suitable for IF staining were found (19 peptide/fragment antibodies with defined sequence, 3 peptide/fragment antibodies with undefined sequence, 4 antibodies against full length protein sequence, 2 undefined antibodies (proprietary to its manufacturer)). All undefined antibodies were excluded from further selection. **From 23 antibodies with a defined sequence, we selected two representatives: #1 (Abcam, ab201683) recently used in various super-resolution studies addressing replisome dynamics (Lee et al, Nat Commun, 2021, Lee et al, Methods Enzymol, 2021, Yin et al, Mol Cell, 2021), and #2 (Novus Biologicals, H00004175-M04).**

Rebuttal Fig. 5. A graphical summary of MCM6 antibody selection. a) Definition of antigen sequence in MCM6 collection of antibodies suitable for IF. b) A table of MCM6 antibodies with defined sequence including information on host, clonality and number of citations according to AB cite online tool. c) Antigen localization of MCM6 antibodies listed in (b).

For MCM7 (Rebuttal Fig. 6), 58 antibodies suitable for IF staining were found (33 peptide/fragment antibodies with defined sequence, 8 peptide/fragment antibodies with undefined sequence, 10 antibodies against full length protein sequence, 6 undefined antibodies (proprietary to its manufacturer)). All undefined antibodies were excluded from further selection. **From 43 antibodies with a defined sequence, we selected one representative: #1 (Santa Cruz, sc-9966), which is raised against full length of MCM7 and has the highest citation rate among all MCM7 antibodies. Based on the high citation rate, we consider this antibody as a field-accepted antibody and used it as a reference throughout this manuscript.**

Rebuttal Fig. 6. A graphical summary of MCM7 antibody selection. a) Definition of antigen sequence in MCM7 collection of antibodies suitable for IF. b) A table of MCM7 antibodies with defined sequence including information on host, clonality and number of citations according to AB cite online tool. c) Antigen localization of MCM7 antibodies listed in (b).

- It would be useful for the reader if the authors could highlight the position of the Mcm4-Halo and Mcm2-GFP tags on a replisome structure, analogous to the depictions in Figure 2e, f.

Due to the inherent flexibility of the linkers connecting the C-termini of MCM2 and MCM4 and their respective tags, we are only able to provide a rough estimate of their relative positions for the Reviewer (Rebuttal Fig. 7). The linker lengths and thus the distances used for generating the figure are based on the total number of unstructured amino acids between last structured C-terminal residue of the MCMs (MCM2 I900/904 aa; MCM4 R797/863 aa) in the recent cryo-EM structure of the human core replisome (PDB: 7PFO, Jones et al., EMBO J, 2021) and the first structured residues in the GFP/Halo structures. This length includes the flexible linkers designed into the sequence (22 aa between MCM2 and GFP; 37 aa between MCM4 and Halo) and, in total, is 29 aa for MCM2-GFP and 104 aa for MCM4-Halo. Using an approximation of 3 Å per amino acid (a fully extended B-sheet would be ~3.5 Å per aa), we can estimate the flexibility and an

approximate radius for how far the tags might extend from the core replisome. We depict the uncertainty in the location by the coloured areas in the replisome structures.

Rebuttal Fig. 7 A graphical depiction of the possible positions of Halo- and GFP-tags in relation to the core replisome structure. Left, a graphical representation of the human core replisome with Halo-tagged MCM4. Right, a graphical representation of the human core replisome with GFP-tagged MCM2. The colored areas represent the uncertainty in the potential positions of the GFP and Halo tags due to the length and flexibility of their respective linkers.

- It would be nice if the authors could be more systematic in their approach (also relevant to the above point about antibody selection). For example, does siRNA against other replisome proteins (e.g. CLASPIN, AND-1) allow detection of Mcm2-7 at replication forks, or did the authors only try TIMELESS depletion?

This is again a very good suggestion by the Reviewer, which aligns well with a similar comment raised by Reviewer #1. We kindly ask this Reviewer to see our detailed response to this issue on p. 2 (bottom) and p. 3 in this rebuttal. To briefly summarize our conclusions also here, we now expanded the previous data with TIMELESS by including depletions of CLASPIN and AND-1 (Supplementary Fig. 9b-k). Aligned with our structural analysis, we observed that CLASPIN depletion unmasks MCM2 and MCM6 subunits leading to increased detection of CDC45-marked RFs (Supplementary Fig. 9b-d), while epitopes of MCM7 subunit located opposite the CLASPIN interaction interface remained constant (Supplementary Fig. 9e). Importantly, we showed that AND-1 depletion had no effect on colocalization of CDC45 with various MCM antibodies because the AND-1 interaction with CMG is mediated primarily through CDC45 and GINS and not MCMs (Supplementary Fig. 9g-j). Taken together, our data suggest that the active MCM scaffold is coated with multiple replisome factors that limit antibody access during IF staining. Moreover, our approach provides important in-cell validation of the position of individual RPC components in the human replisome structure.

Also, did the authors try tagging other Mcm2-7 subunits in addition to Mcm2 and 4? Given the technical nature of the paper, this type of information would be very useful.

Since we showed that endogenous tagging of two different MCM subunits with different fluorescent tags (Halo or GFP tag) can overcome the steric hindrance of MCM antibody epitopes

in the replisome structure, we did not perform endogenous tagging of additional MCM subunits which is time- and cost-consuming procedure to generate such targeting constructs and cell line. Still, to address this Reviewer's comment, which makes an important point about the general implications of our results, we used the targeting constructs from the original submission to endogenously tag the MCM4 subunit in immortalized diploid RPE-1 cells. Gratifyingly, we could fully reproduce our previous results from U2OS cells by showing that also in RPE-1 cells, MCM4-Halo (but not immunostained MCM7 used here as a reference antibody) strongly co-localized with active RFs marked by PCNA (Supplementary Fig. 6b).

- It seems pretty clear that the differences in the ability of the anti-MCM and anti-tag antibodies to detect MCM2-7 as part of CMG is not due to intrinsic differences in the sensitivities of the various antibodies (for example, the TIMELESS data in Figure 4 argues against this). This fact should be stated for clarity.

This is indeed an excellent point, which motivated us to design an experiment, in which we compared the fluorescent intensity of all MCM antibodies and endogenously tagged MCM4 under the same imaging condition. Apart from directly addressing the Reviewer's point, these QIBC data also confirmed that even the MCM6 #2 antibody, which as the only one from the extended antibody selection with a limited potential to decorate large replication factories, was not biased by different antibody sensitivity (Supplementary Fig. 7a, b).

- The fact that TIMELESS depletion enhances the co-localised signal between endogenous MCM2-7 and Cdc45 but not HALO-tagged MCM4 and Cdc45 is an informative control, as it shows that the TIMELESS effect is most likely due to steric occlusion of MCM2-7 epitopes rather than, for example, any impact upon DNA replication per se, or replisome disassembly (TIMELESS is required for optimal replisome disassembly in worms – Xia et al, EMBO J, 2021). Again, a statement on this point would be helpful for clarity.

We thank the Reviewer for this helpful comment. We have now included data showing that none of the RPC depletion affected the colocalization between endogenously tagged MCM4-Halo and CDC45 (Fig. 4e, Supplementary Fig. 9f, k), indicating that the increased Pearson correlation upon TIMELESS or CLASPIN depletions (Fig. 4a-c, Supplementary Fig. 9b-d) is indeed caused by the unmasking of active MCMs in RFs rather than any adverse effect on DNA replication in general.

Reviewer #3 (Remarks to the Author):

In the manuscript: "Solving the MCM paradox by visualizing the MCM scaffold of endogenous CMG helicase at active replisomes" by Dr Polasek-Sedlackova and others, the authors revisit the so called MCM paradox i.e the inability to detect MCMs at active replication forks by immunofluorescence despite it having an essential function in replication elongation. In part the reason behind the paradox is existence of many folds higher number of inactive MCM2-7 dormant origins that are positioned in yet unreplicated parts of the genome and can mask the detection of MCM2-7 within active replisomes. Through population based quantitative immunofluorescence (QIBC) and CRISPR aided endogenous tagging of MCMs, the authors concluded that the steric hindrance caused during conversion of MCM-DHs to active CMGs leads to epitope masking thereby causing limited detection by antibody-based immunofluorescence (IF). Despite having done very decisive and clear experiments to prove their claim, the manuscript seems more suitable as being titled as a methodological development in detection of MCMs at active replication forks rather than reporting any novel biological phenomenon/activity. In summary, it is a nice study with robust experiments, but it does seem to bring very limited novel biological discovery.

We sincerely thank the Reviewer for appreciating our work. Although our work provides a simple

explanation of the longstanding MCM paradox, we believe it is an important missing piece of information for a better understanding of replisome architecture and its dynamics in living cells.

Major points

1. Authors set up the scene as if there were doubts whether MCM2-7 form part of a replicative helicase (“Are MCM proteins parts of functional replisomes in cells?”, line 52). However, in my mind there is no such doubt in the literature at all, as many studies isolated MCM2-7 as part of active replisomes or shown their role in DNA unwinding in different ways (e.g. Labib et al.,2000; Calzada et al.,2005; Gambus et al.,2006; Pacek et al.,2006; Mašata et al.,2011; Labib et al.,2017).

We agree with the Reviewer that a huge wealth of studies support the notion that the MCM2-7 complex is a part of the replisome and acts as the main replicative helicase across multiple eukaryotic species. However, to make a definite explanation of the MCM paradox, we are bound to also test the hypothesis that considers an alternative replisome architecture in cells (*Laskey & Madine, EMBO Rep, 2003*) even though it seems to be less likely in light of many structural and biochemical studies. Ultimately, until our successful visualization of CMG helicase (via endogenous tagging of CDC45 and MCMs) at the RFs, we did not know whether the longstanding MCM paradox was a biological or technical phenomenon. Our study provides a clarification of this conundrum, which we believe is definitive, yet surprisingly simple (and thus approachable by many labs). We trust that the insight and tools provided by our work have a strong potential to bring structural and cell biologists closer than ever to cross-examine the key concepts defining the dynamics of DNA replication and its response to the constantly changing cellular environment.

2. Authors conclude that limited detection of MCM2-7 within active replisomes by antibodies is due to the fact that their epitopes are masked by other replisome components and so these antibodies are much better at detecting inactive MCM2-7 than the ones within the large replisome machinery. It would be however nice if more than one antibody was tested for each of the MCM subunits. In fact, most of the antibodies used in this study (except for MCM5) are monoclonal antibodies or polyclonal antibodies against a short synthetic peptide. In each case the antibody detects only a very small epitope within the MCM protein and so indeed can be affected by interacting partners of this protein. It would be ideal to test a catalogue of polyclonal antibodies raised against large chunks of the antigen to really prove that MCMs within replisomes are undetectable by antibody based IF.

This is an important point, which was also raised by Reviewer #2 and where we, therefore, invested a lot of effort to strengthen this part. We kindly ask this Reviewer to read our detailed response on pp. 4-8 in this rebuttal. Here we provide the summary of the manuscript amendments in this regard and highlight a few additional pieces of information, which we hope the reviewers find informative.

In a nutshell, we have now tested a much-extended panel of MCM antibodies to validate our main conclusions. To help the field as much as possible, we selected antibodies based on the number of citations according to the cite AB online tool. Additionally, we considered the location and the length of the antibody antigen as well as the presence of monoclonal and polyclonal antibody types in our catalogue. Whenever possible, we always included antibodies raised against full length protein sequence or the largest antigen, if we were unable to find commercially available antibodies against full length suitable for IF. Detailed information on antibodies has now been included in the new paragraph in the method section (replacing the previous Supplementary Table 1 and including much more detail).

We also want to highlight that from all MCM antibodies we identified one mouse monoclonal full-length MCM6 #2 with 0 citations as the only antibody partially detecting active MCMs in RFs (see Fig. 2d-f, Supplementary Fig. 4b, Supplementary Fig. 5a, b), which is in sharp contrast to mouse monoclonal full-length MCM7 used in the field as the “gold standard” with more than 150

citations. QIBC and colocalization analysis consistently support the conclusion that in contrast to endogenously tagged MCMs (Supplementary Fig. 6c-e), the MCM6 #2 antibody detects RFs only partially due to the steric shielding of the MCM6 subunit by CLASPIN (Fig. 2h, Supplementary Fig. 9d). Taken together, our new data further expand and strengthen our original conclusion that active MCMs as part of the replication fork are shielded by the replisome components and are therefore largely undetectable by the commonly available antibodies.

3. Others also used tagged MCM2-7 (e.g. Symeonidou et al. 2013) and come to a different conclusion – their failed to detect a clear interaction of tagged MCM2-7 with sites of replication. Why the discrepancy?

This is a very important comment raised by the Reviewer. We have now included the data showing that uncontrolled MCM overexpression without removal of endogenous pool of MCMs leads to the production of MCM2-7 complexes which behave as inactive replication origins (Supplementary Fig. 7c-f). Our new data showed that ectopically overexpressed MCMs possess a very low ability to interact with CDC45 in direct comparison to endogenously tagged MCMs (Supplementary Fig. 7f). Although authors in previous papers made great efforts to ectopically overexpress MCMs at a physiological level (Kuipers et al, *JCB*, 2011, Symeonidou et al, *JBC*, 2013), untagged endogenous MCMs were not removed and therefore colocalization analysis to detect MCMs in RFs was not successful.

Please also see also our response to the related comment by Reviewer #1 (p. 2 in this rebuttal), who raised similar question, and where we discuss additional important ramifications of MCM overexpression.

4. The authors show that detection of endogenously tagged MCMs or CDC45 can overcome the problems of using epitope specific antibodies and detect MCM2-7 within the active replisome. However, protein tagging, especially with large size GFP tag, comes with its own problems – as authors point out themselves these are tight protein complexes that are changing dynamically during the cell cycle and large tags on some of their components can change complex dynamics and formation. I think it is essential that authors verify cell proliferation and complex interactions in cells with tagged MCMs and CDC45.

This is a valid point, which we thoroughly addressed (including immunoprecipitation experiments of endogenously tagged MCM2, MCM4, and CDC45, exactly as suggested by the Reviewer) in our previous paper (Sedlackova et al., *Nature*, 2020). We thank the Reviewer for reminding us that we did not provide a clear link to this set of controls, which we now rectified in the revised by including the reference to our paper in the Methods section. Importantly, we could show that all tagging and gene editing reported in these studies did not affect cell proliferation and preserved interaction with partners as untagged endogenous proteins.

5. Replisome disassembly is prevented by the presence of the lagging strand during replication elongation and not by masking of the replisome subunits during replication termination. Therefore, preventing replisome disassembly may not be the best example of the physiological relevance of the masking of MCMs caused during conversion to active CMGs.

We initially wanted to point out that the MCM scaffold might be hidden not only by the replisome factors but also by DNA such as the presence of a lagging strand, the loss of which during replication termination leads to the exposure of the LRR1 binding site. However, we agree with the Reviewer that this might not be the best example. Therefore, we have now removed this statement from the revised manuscript. Nevertheless, we expect that studying the regulation of MCM scaffold exposure as a mechanism of replication fork remodeling will be an important area for our future investigation, fully harnessing the tools that we generated during this study.

Minor points:

1. In Fig. 3F middle panel, the Pearson correlation coefficient at S1 and S2 do not vary significantly, contradicting their representative images in Fig. 3E. A similar problem exists with EDF. 6A and B.

We do not observe major changes in the Pearson correlation between CDC45 and MCMs in the early stages of the S phase due to the large fraction of inactive MCMs present on the chromatin. Although the CDC45 level on chromatin is changing in these early stages, Pearson correlation coefficient calculations over the determined threshold are independent of CDC45 levels. Moreover, in this stage of the S phase DNA synthesis occur in more dispersed replication sites compared to late-replicating discrete replication foci, which might also contribute to the sensitivity of colocalization analysis. Stimulated by this comment, we will deposit raw images in an open database, where readers may browse all acquired images.

2. Page 4: line 123: “increased retention” should be rephrased as “increased detection”.

We have incorporated this change in the revised manuscript.

3. The table listing epitope sequence of antibodies should be renamed as antigen sequence.

In the revised manuscript, we removed Supplementary table 1 as we provide detailed information about MCM antibodies in the method section. However, we used the suggested nomenclature by the Reviewer in the new paragraph describing the selection of MCM antibodies.

4. The range of CDC45-GFP signals in unperturbed conditions (DMSO) in Fig. 2A is very low despite being identical to conditions in Fig 1C, and EDF. 1B. In fact, the increased signals upon ATR inhibition seem more like the unperturbed ones in Fig 1C, and EDF. 1B.

In Fig. 1c and Supplementary Fig. 1b, the exposure time for the 488 channel was 500 ms. In Fig. 2a, due to the ATR inhibitor treatment which increases chromatin binding of CDC45-GFP approximately four-fold (represented by increased fluorescent intensity), we modified the imaging conditions for the 488 channel by decreasing exposure time to 150 ms in order to avoid saturated pixels in our acquired images. Therefore, the fluorescent intensity of CDC45 chromatin binding appears very low in the DMSO condition. Stimulated by the Reviewer’s comment, we have now updated a paragraph about QIBC settings in the method section and included information about exposure times for each experiment in the source data for QIBC.

5. It is not clear why in EDF. 4C and Fig 3: A-F a particular endogenous MCM subunit is compared to the tagged version of a different MCM subunit (MCM7 vs MCM4-Halo/MCM3 vs MCM2-GFP).

We used MCM7 as a reference MCM antibody throughout the manuscript because we consider this antibody as a field-accepted antibody to visualize the MCMs (having more than 150 citations according to the cite AB online tool). Therefore, when studying opposite chromatin kinetics of immunostained and endogenously tagged MCMs, we showed in the main figures (Fig. 3a-c) data for the MCM7 antibody and in Supplementary figures, we use the MCM4 antibody to compare the same MCM subunit (Supplementary Fig. 5c). However, while revising our manuscript, we noticed that former figures Extended Data Fig. 4c and Extended Data Fig. 5b comparing chromatin kinetics of endogenously tagged MCM2-GFP and immunostained MCM3 are additional repetitions of data comparing the same subunit. Thus, to avoid redundancy and save space, we decided to consistently use MCM7 antibody throughout the manuscript and remove the above-mentioned figures comparing endogenously tagged MCM2-GFP and immunostained MCM3.

6. Page 3: line 97-99: If the MCM5 epitope is involved in MCM2-7 dimerization, it is expected to be masked from antibody recognition in double hexamers as much as in CMG. It is therefore not expected to be detected by IF both in MCM-DHs as well as active CMG configurations.

By describing the position of the MCM5 epitope in double hexamers and replisome structures we wanted to point out that different conformations of the epitope may exist for different biologically functional complexes. We agree with the Reviewer that one would expect that the MCM5 antibody would not detect MCM double hexamer as well. However, this was not confirmed by our IF imaging. One of the potential explanations could be fine differences in structures of *S. cerevisiae* (referred to in our study) and human MCM double hexamers (structures of which have not yet been reported). Until the structure of the human MCM double hexamer is solved we can only speculate. To avoid any overinterpretation, we decided to remove this statement. In the revised manuscript, we omit detailed information about individual antibody epitopes and describe the inaccessibility of active MCMs in the replisome structure in a general way, because we now provide several examples of full length MCM antibodies that are not detecting RFs (except partial detection by MCM6 #2 antibody as detailed above).

7. While the authors claim the use of CDC45 at active RFs as a proxy to distinguish between the bulk of MCMs from the active ones at the RFs as a novel approach, colocalization with PCNA and EdU are equally good proxy indicators used in several studies in the past.

We agree with the Reviewer that PCNA and EdU are good proxies to mark replication factories. However, we think that in future studies CMG helicase or leading strand polymerase may serve as better proxies for studying molecular changes in replisome architecture by super-resolution microscopy or nanoscopy techniques.

References included in the Reviewer's comments:

- Calzada et al., Molecular anatomy and regulation of a stable replisome at a paused eukaryotic DNA replication fork. *Genes Dev* (2005)
- Gambus et al., GINS maintains association of Cdc45 with MCM in replisome progression complexes at eukaryotic DNA replication forks. *Nat Cell Biol* (2006)
- Labib et al., Uninterrupted MCM2-7 Function Required for DNA Replication Fork Progression. *Science* (2000).
- Labib et al., MCM2–7 Proteins Are Essential Components of Prereplicative Complexes that Accumulate Cooperatively in the Nucleus during G1-phase and Are Required to Establish, But Not Maintain, the S-phase Checkpoint. *Molecular Biology of the Cell* (2017)
- Mašata et al., A fraction of MCM 2 proteins remain associated with replication foci during a major part of S phase. *Folia Biologica (Praha)* (2011)
- Pacek et al., Localization of MCM2-7, Cdc45, and GINS to the site of DNA unwinding during eukaryotic DNA replication. *Mol. Cell* (2006)
- Symeonidou et al., Multi-step Loading of Human Minichromosome Maintenance Proteins in Live Human Cells. *Journal of Biological Chemistry* (2013)

References included in answers to Reviewer's comments:

- Jones, M. L., Baris, Y., Taylor, M. R. G. & Yeeles, J. T. P. Structure of a human replisome shows the organisation and interactions of a DNA replication machine. *EMBO J* 40, e108819, doi:10.15252/embj.2021108819 (2021).**
- Laskey, R. A. & Madine, M. A. A rotary pumping model for helicase function of MCM proteins at a distance from replication forks. *EMBO Rep* 4, 26-30, doi:10.1038/sj.embor.embor706 (2003).**

- Lee, W. T. C., Gupta, D. & Rothenberg, E. Single-molecule imaging of replication fork conflicts at genomic DNA G4 structures in human cells. *Methods Enzymol* 661, 77-94, doi:10.1016/bs.mie.2021.08.008 (2021).
- Lee, W. T. C. et al. Single-molecule imaging reveals replication fork coupled formation of G-quadruplex structures hinders local replication stress signaling. *Nat Commun* 12, 2525, doi:10.1038/s41467-021-22830-9 (2021).
- Sedlackova, H. *et al.* Equilibrium between nascent and parental MCM proteins protects replicating genomes. *Nature* 587, 297-302, doi:10.1038/s41586-020-2842-3 (2020).
- Somyajit, K. et al. Redox-sensitive alteration of replisome architecture safeguards genome integrity. *Science* 358, 797-802, doi:10.1126/science.aao3172 (2017).
- Yin, Y. et al. A basal-level activity of ATR links replication fork surveillance and stress response. *Mol Cell* 81, 4243-4257 e4246, doi:10.1016/j.molcel.2021.08.009 (2021).

REVIEWERS' COMMENTS

Reviewer #1 (Remarks to the Author):

The authors have done a considerable amount of additional work to characterize anti-MCM antibodies for the scientific community and - in response to one of my criticisms - knocking down additional replication proteins with varied potential to mask MCM epitopes. Their response to the critique of the literature using tagged MCM in living cells, assuming they have all the papers (I will trust that they do), is just a bit less than fully convincing but they have addressed it as well as is reasonable. There are still issues of different cell lines, and whether chronic "as low as possible" ectopic expression of tagged MCM (which also lowers endogenous protein expression) chronically (cycle after cycle) favors use of the untagged MCMs (wasn't quite convinced of this argument because tagged MCM must eventually get into the pool of "old" MCMs) and I'm not sure it entirely rules out a role for dilution of signal upon de-condensation of chromatin since in the cited paper, histone signal was also decreases at replication foci, and surely histones are not removed en masse. HOWEVER - the authors have raised and dealt with this concern and have clearly demonstrated that epitope masking is a highly likely explanation, if not THE explanation, for the paradox. As to whether this work is a technical footnote, as the other reviewers suggest, they are correct that no one doubts that MCM is the replicative helicase but this issue has been a lingering headache for the field. NC is a high volume journal and this is solid work, as all 3 reviewers agree. Moreover, epitope masking is not restricted to DNA replication - such a headache is a threat to all sorts of cell biological applications but rarely does anyone investigate it so thoroughly. I think this work will be helpful to anyone studying large macromolecular biological complexes.

Minor point- Supplementary Figure 7f first panel lacks the label.

Reviewer #2 (Remarks to the Author):

The authors have rigorously addressed all the comments raised in the first review, resulting in a very thorough and focussed manuscript that will be of use and interest to the DNA replication and related fields. I fully support publication of this work in Nature Communications.

Reviewer #3 (Remarks to the Author):

The authors done a very good job and responded very well to all comments from reviewers. I do not have any more comments as my previous questions were robustly addressed.

POINT-BY-POINT RESPONSE TO THE REVIEWER'S COMMENTS

We would like to thank all three reviewers for their enthusiasm, valuable comments, and excellent guidance during the revision process.

Reviewer #1 (*Remarks to the Author*):

The authors have done a considerable amount of additional work to characterize anti-MCM antibodies for the scientific community and - in response to one of my criticisms - knocking down additional replication proteins with varied potential to mask MCM epitopes. Their response to the critique of the literature using tagged MCM in living cells, assuming they have all the papers (I will trust that they do), is just a bit less than fully convincing but they have addressed it as well as is reasonable. There are still issues of different cell lines, and whether chronic "as low as possible" ectopic expression of tagged MCM (which also lowers endogenous protein expression) chronically (cycle after cycle) favors use of the untagged MCMs (wasn't quite convinced of this argument because tagged MCM must eventually get into the pool of "old" MCMs) and I'm not sure it entirely rules out a role for dilution of signal upon de-condensation of chromatin since in the cited paper, histone signal was also decreases at replication foci, and surely histones are not removed en masse. HOWEVER - the authors have raised and dealt with this concern and have clearly demonstrated that epitope masking is a highly likely explanation, if not THE explanation, for the paradox. As to whether this work is a technical footnote, as the other reviewers suggest, they are correct that no one doubts that MCM is the replicative helicase but this issue has been a lingering headache for the field. NC is a high volume journal and this is solid work, as all 3 reviewers agree. Moreover, epitope masking is not restricted to DNA replication - such a headache is a threat to all sorts of cell biological applications but rarely does anyone investigate it so thoroughly. I think this work will be helpful to anyone studying large macromolecular biological complexes.

Our response:

We sincerely thank the Reviewer for his/her appreciation for our work and for additional textual suggestions, which are again very helpful. We agree with the reviewer that there is a variety of model systems using tagged MCMs in the current literature. We now specify (page 5, line 158) that we are discussing our results specifically in the context of previous papers using mammalian cells (within these experimental systems, we indeed believe that our literature coverage is complete). We also agree with the reviewer that chronic ("as low as possible") expression of ectopic MCMs driven by heterologous promoters, which concurrently partially lowers endogenous protein expression levels, might potentially increase the engagement of MCMs in active replisomes. At the same time, we caution that such conditions would be by nature very variable from experiment to experiment (and from lab to lab) and thus difficult to reproduce. Given the fact that only a very small fraction (5-10%) of parental MCMs are converted to active CMG helicases would further preclude generating a 'clean' experimental system based on MCM overexpression. Indeed, as we showed in our earlier work (*Sedlackova et al., Nature 587, 297-302, 2020*) the MCM equilibrium is exquisitely sensitive to the exact stoichiometry of its 6 subunits. Combined with our current manuscript, we believe that the only rigorous way to bypass epitope masking and visualize the entire pool of active MCMs in their natural settings is endogenous tagging. This key conclusion is clearly summarized in the manuscript (p. 5, lines 174-178) as follows: "Consequently, detection of MCMs at RFs will be challenging in all studies that ectopically express MCMs without removing the cycling endogenous population. Collectively, we conclude that CRISPR-Cas9 endogenous tagging of MCM subunits effectively bypasses the steric inaccessibility of MCMs in fully assembled CMGs and might be the only robust and reliable way to visualize the active pool of MCMs in cells". Finally, we would like to thank the reviewer for the extremely nice and encouraging 'conceptual bonus' to the above reasoning by concluding his/her final comment by stating that: "epitope masking is not restricted to DNA replication - such a headache is a threat to all sorts of cell biological applications but rarely does anyone investigate it so thoroughly. I think this work will be helpful to anyone studying large macromolecular biological complexes". We were really delighted to hear from the reviewer what

we were also aiming at, namely inspiring the readers beyond the DNA replication community.

Minor point- Supplementary Figure 7f first panel lacks the label.

We have now included the label in Supplementary Fig. 7f.

Reviewer #2 (Remarks to the Author):

The authors have rigorously addressed all the comments raised in the first review, resulting in a very thorough and focused manuscript that will be of use and interest to the DNA replication and related fields. I fully support publication of this work in Nature Communications.

Our response:

This Reviewer is satisfied with our revision and does not have additional specific comments. We are very grateful for his/her continued enthusiasm for our work and inspiring input throughout the reviewing process.

Reviewer #3 (Remarks to the Author):

The authors done a very good job and responded very well to all comments from reviewers. I do not have any more comments as my previous questions were robustly addressed.

Our response:

This Reviewer is also satisfied with the way how we developed our manuscript. We are again very grateful for his/her insightful suggestions throughout the reviewing process.